

# Long-distance entanglement in Motzkin and Fredkin spin chains

**Luca Dell'Anna**

Dipartimento di Fisica e Astronomia "G. Galilei", Università di Padova,
via F. Marzolo 8, I-35131, Padova, Italy

## Abstract

We derive some entanglement properties of the ground states for two classes of quantum spin chains described by the Fredkin model, for half-integer spins, and the Motzkin model, for integer ones. Since the ground states of the two models are known analytically, we can calculate exactly the entanglement entropy, the negativity and the quantum mutual information. We show, in particular, that these systems exhibit long-distance entanglement, namely two disjoint regions of the chains remain entangled even when their separation is sent to infinity, i.e. these systems are not affected by decoherence. This strongly entangled behavior, occurring both for colorful versions of the models (with spin larger than 1/2 or 1, respectively) and for colorless ones (with spin 1/2 and 1), is consistent with the violation of the cluster decomposition property. Finally, we show that this behavior involves disjoint segments located both at the edges and in the bulk of the chains.

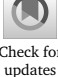

# 1   Introduction

The study of nonlocal properties and their consequences on the dynamics in addition to the violation of the area law for the entanglement entropy are certainly, at the present date, a very challenging field of research. The concept of locality plays a crucial role in physical theories, with far reaching consequences, a fundamental one being the cluster decomposition property [1, 2]. This property implies that two-point connected correlation functions go to zero when the separation of the points goes to infinity. This is the reason why two systems very far apart, separated by a large distance, behave independently.

Another aspect related to correlations is the quantum entanglement. In bipartite systems the von Neumann or entanglement entropy quantifies how the two parts of the whole system in a quantum state are entangled. This quantity measures non-local quantum correlations and has universal properties, like the fact that, for gapped systems in the ground states, it scales with the area of the boundary of the two subsystems [3]. This property is called area law and is valid for systems with short-range interactions. In other words, if the interactions are short-ranged the information among the constituents of the system propagates with a finite speed involving a surface surrounding the source of the signal, like an electromagnetic impulse propagating with the speed of light. For critical one-dimensional short-range systems the area law is violated logarithmically [4]. Quantum spin chains are promising tools for universal quantum computation [5] and the efficiency may be related to the amount of quantum entanglement. Spin systems with entanglement entropy larger than that dictated by the area law can be used for quantum computing even more efficiently, and breaking down the speed of the propagation of the excitations can represent a breakthrough for quantum information processing.

Recently, novel quantum spin models have been introduced, with integer [6–8] (Motzkin model) and half-integer [9, 10] spins (Fredkin model), which, in spite of being described by local Hamiltonians, exhibit violation of the cluster decomposition property and of the area law for the entanglement entropy, with the presence of anomalous and extremely fast propagation of the excitations after driving the system out-of-equilibrium [9]. These models seem, therefore, extremely promising for applications in quantum information and communication processes. Very recently, also deformed versions of Motzkin [11] and of Fredkin [12] chains have been introduced and studied [13, 14], which can exhibit a quantum phase transition separating an extensively entangled phase [11, 12] from a topological one [15].

Quite recently, it has been given also a continuum description for the ground-state wave-functions of those models, as originally formulated and in the colorless cases (spin-1 Motzkin model and spin-1/2 Fredkin model), which can reproduce well some quantities like the local magnetization and the entanglement entropy, and whose scaling Hamiltonian is not conformally invariant [16]. Some results on the Renyi entropy for these models have been also reported [17].

In this work we focus on the study of quantum entanglement after the discovery that cluster decomposition property in such systems can be violated [9]. This behavior has been shown to occur in the connected correlation functions of spins along $z$-directions for the colorful cases (for spins larger than 1), when also the area law for the entanglement entropy is violated more than logarithmically [7, 9], and is more pronounced for correlation functions measured close to the edges of the chains. What is presented here is the calculation of other entanglement measures, the quantum negativity [18, 19] and the mutual information [20, 21] shared by two disjoint segments of the chains in the ground state. The latter, also called the von Neumann mutual information, is a measure of correlation between subsystems of a quantum state and is the quantum analog of the Shannon mutual information. The negativity is also a measure of quantum entanglement, based on the positive partial transpose (PPT) criterion of separability, which provides a necessary condition for a joint density matrix $\rho_{AB}$ of two quantum mechanical systems, $A$ and $B$, to be separable [18]. If the state is separable (not entangled) the negativity is zero, therefore, if the negativity is greater than zero, the state is surely entangled, nevertheless it can be zero even if the state is entangled.

We show that such systems exhibit long-distance entanglement [22], namely given a measure of entanglement, e.g. the mutual information, $\mathcal{I}_{AB}$ for the system $A \cup B$ made by two disjoint subsystems $A$ and $B$, the quantity $\mathcal{I}_{AB}$ does not vanish when the distance between $A$ and $B$ goes to infinity. This result is consistent with the fact that also some connected correlation functions do not vanish in the thermodynamic limit [9, 16, 23], being the mutual information an upper bound for normalized connected correlators [21, 24].

Moreover we show that this non-vanishing mutual information, persisting for infinite distances, is verified not only in the colorful cases, where the entanglement entropy scales as a square-root law, but also in the colorless cases, where the entanglement entropy scales just logarithmically, as for critical systems. Finally, contrary to what found in the continuum limit [16], this behavior occurs, and is even more pronounced, also when the subsystems are located deep inside the bulk, showing a stronger entanglement as compared to the one obtained in conformal field theories, where the mutual information vanishes upon increasing the distance with a power law behavior [25]. On the other hand, quite surprisingly, we show that the mutual information for two disjoint subsystems inside the bulk has the same form of the logarithmic negativity and of the mutual information for conformal field theories of two adjacent intervals [26].

## 2 Models

In this section we report the Hamiltonians for recently introduced half-integer and integer spin models, referred to as Fredkin and Motzkin models, whose ground states are known exactly and have the peculiarity of being related to some random lattice walks.

## 2.1 Fredkin model

The Fredkin model [9, 10] is described by the following half-integer spin Hamiltonian $H = H_0 + H_\partial$, with

$$H_0 = \sum_{c,\bar{c}=1}^{q} \left\{ \sum_{j=1}^{L-2} \left[ \mathsf{P}\left( \left| Q_{\uparrow j}^{c\bar{c}} \right\rangle \right) + \mathsf{P}\left( \left| Q_{\downarrow j}^{c\bar{c}} \right\rangle \right) \right] + \sum_{j=1}^{L-1} \mathsf{P}\left( \left| Q_{0j}^{c\bar{c}} \right\rangle \right) \right\} + \sum_{c \neq \bar{c}}^{q} \sum_{j=1}^{L-1} \mathsf{P}\left( \left| \uparrow_j^c \downarrow_{j+1}^{\bar{c}} \right\rangle \right), \quad (1)$$

$$H_\partial = \sum_{c=1}^{q} \left[ \mathsf{P}\left( \left| \downarrow_1^c \right\rangle \right) + \mathsf{P}\left( \left| \uparrow_L^c \right\rangle \right) \right] \quad (2)$$

composed by bulk, $H_0$, and boundary, $H_\partial$, Hamiltonians, where $\mathsf{P}(|.\rangle) = |.\rangle\langle.|$ is a projector operator acting on quantum states made by local spin-states, $|\uparrow_j^c\rangle$, located on sites labelled by $j$ with half-integer spins along $z$-quantization axis $s_z = \left( c - \frac{1}{2} \right)$ and $|\downarrow_j^c\rangle$ with local half-integer spins $s_z = \left( \frac{1}{2} - c \right)$, with $c \in \mathbb{N}$ and $c = 1, \ldots, q$. The maximum value of the index $c$ is $q$ that is called the number of *colors* of the model. The quantum states appearing in Eq. (1) are defined by

$$\left| Q_{\uparrow j}^{c\bar{c}} \right\rangle = \frac{1}{\sqrt{2}} \left( \left| \uparrow_j^{\bar{c}} \uparrow_{j+1}^c \downarrow_{j+2}^c \right\rangle - \left| \uparrow_j^c \downarrow_{j+1}^c \uparrow_{j+2}^{\bar{c}} \right\rangle \right), \quad (3)$$

$$\left| Q_{\downarrow j}^{c\bar{c}} \right\rangle = \frac{1}{\sqrt{2}} \left( \left| \downarrow_j^{\bar{c}} \uparrow_{j+1}^c \downarrow_{j+2}^c \right\rangle - \left| \uparrow_j^c \downarrow_{j+1}^c \downarrow_{j+2}^{\bar{c}} \right\rangle \right), \quad (4)$$

$$\left| Q_{0j}^{c\bar{c}} \right\rangle = \frac{1}{\sqrt{2}} \left( \left| \uparrow_j^c \downarrow_{j+1}^c \right\rangle - \left| \uparrow_j^{\bar{c}} \downarrow_{j+1}^{\bar{c}} \right\rangle \right). \quad (5)$$

For colorless case, $q = 1$ (spin $1/2$), we have that the third and the last term, so-called crossing term, in Eq. (1) are not present. In this case the bulk Hamiltonian, in terms of Pauli matrices, reads

$$H_0 = \sum_{j=1}^{L-2} \left[ (1 + \sigma_{z,j})(1 - \vec{\sigma}_{j+1} \cdot \vec{\sigma}_{j+2}) + (1 - \vec{\sigma}_j \cdot \vec{\sigma}_{j+1})(1 - \sigma_{z,j+2}) \right]. \quad (6)$$

In terms of Fredkin gates $\hat{F}_{ijk}$ (controlled-swap operators), the Hamiltonian in Eq. (6) becomes $H_0 = \sum_j^{L-2} (2 - \hat{F}_{j,j+1,j+2} - \sigma_{j+2}^x \hat{F}_{j+2,j+1,j} \sigma_{j+2}^x)$, where $\hat{F}_{i,j,k}$ acts on three $\frac{1}{2}$-spins (three qbits), swapping the $j$-th and $k$-th spins if the $i$-th is in the state $|\uparrow\rangle$ while does nothing if it is in the state $|\downarrow\rangle$.

## 2.2 Motzkin model

The Motzkin model [6,7] is described by the following integer spin Hamiltonian $H = H_0 + H_\partial$ with

$$H_0 = \sum_{c=1}^{q} \sum_{j=1}^{L-1} \left\{ \mathsf{P}\left( \left| Q_{\Uparrow j}^c \right\rangle \right) + \mathsf{P}\left( \left| Q_{\Downarrow j}^c \right\rangle \right) + \mathsf{P}\left( \left| Q_{0j}^c \right\rangle \right) \right\} + \sum_{c \neq \bar{c}}^{q} \sum_{j=1}^{L-1} \mathsf{P}\left( \left| \Uparrow_j^c \Downarrow_{j+1}^{\bar{c}} \right\rangle \right), \quad (7)$$

$$H_\partial = \sum_{c=1}^{q} \left[ \mathsf{P}\left( \left| \Downarrow_1^c \right\rangle \right) + \mathsf{P}\left( \left| \Uparrow_L^c \right\rangle \right) \right], \quad (8)$$

where now $\mathsf{P}(|.\rangle) = |.\rangle\langle.|$ acts on quantum states made by local integer spin-states, $|\Uparrow_j^c\rangle$ located at sites $j$ with integer spins $s_z = c$, and $|\Downarrow_j^c\rangle$ with spins $s_z = -c$, where, again, $c = 1, \ldots, q$. Also in this case, $q$ is called the number of *colors* of the model and, in the Motzkin case, it corresponds to the maximum value of the spins. The quantum states appearing in Eq. (1) are

defined by

$$\left|Q^c_{\Uparrow j}\right\rangle = \frac{1}{\sqrt{2}}\left(\left|0_j \Uparrow^c_{j+1}\right\rangle - \left|\Uparrow^c_j 0_{j+1}\right\rangle\right), \tag{9}$$

$$\left|Q^c_{\Downarrow j}\right\rangle = \frac{1}{\sqrt{2}}\left(\left|0_j \Downarrow^c_{j+1}\right\rangle - \left|\Downarrow^c_j 0_{j+1}\right\rangle\right), \tag{10}$$

$$\left|Q^c_{0 j}\right\rangle = \frac{1}{\sqrt{2}}\left(\left|0_j 0_{j+1}\right\rangle - \left|\Uparrow^c_j \Downarrow^c_{j+1}\right\rangle\right). \tag{11}$$

For the colorless case, $q = 1$ (spin 1), we have that the last term in Eq. (7) is absent.

## 3 Ground states

The most important property shared by these frustration-free Hamiltonians is that their ground states are unique, made by uniform superpositions of all states corresponding to Motzkin paths, for the integer case (for the Motzkin model) and all states corresponding to Dyck paths for the half-integer one (Fredkin model). These states are such that, denoting the spins up, $\Uparrow$, by /, the spins down, $\Downarrow$, by \ and spins zero, 0, by $-$, one can construct a Motzkin path, while by using only / for $\uparrow$ and \ for $\downarrow$ one can construct a Dick path.

A Motzkin path is any path on a $x$-$y$ plan connecting the origin $(0,0)$ to the point $(0, L)$ with steps $(1,0)$, $(1,1)$, $(1,-1)$, where $L$ is an integer number. Any point $(x, y)$ of the path is such that $x$ and $y$ are not negative.

Analogously, a Dyck path is any path from the point $(0,0)$ to $(0, L)$ (now $L$ should be an even integer number) with steps $(1,1)$, $(1,-1)$. As for the Motzkin path, any point $(x, y)$ of the Dyck path is such that $x$ and $y$ are not negative.

The corresponding colored path are such that the steps can be drawn with more than one color. The color attached to a path move is taken freely only for upward steps (up-spins) while any downward steps (down-spin) should have the same color of the nearest up-spin on the left-hand-side at the same level. This *color matching* is induced by the cost energy contribution described by the last term, both in Eq. (1) and Eq. (7), which, in spite of being short-ranged, it produces non a local effect in the ground state.

As a result, a colorless Motzkin path $|m^{(L)}_p\rangle$ or Dyck path $|d^{(L)}_p\rangle$ can be defined as a string of $L$ spins (or steps) such that, starting from the left by convention, the sum of the spins contained in any initial segment of the string is nonnegative, or alternatively, any initial segment contains at least as many up-spins (upward steps) as down-spins (downward steps), while the sum of all the $L$ spins is zero (the total number of upward steps is equal to the number of downward steps). The colorful Motzkin or Dyck paths are the paths where, in addition, the upward steps can be colored at will while the colors of the downward steps are uniquely determined by the matching condition (any spin down has the same color of the adjacent upward spin on the left-hand-side at the same height). Examples of colored Motzkin and Dyck states are shown in Fig. 1.

The ground state of the Fredkin Hamiltonian is then obtained by a uniform superposition of all possible Dyck paths at a given length $L$ and a given number of colors $q$,

$$|\mathcal{P}^{(L)}\rangle = \frac{1}{\sqrt{\mathcal{D}^{(L)}}} \sum_p |d^{(L)}_p\rangle, \tag{12}$$

where $\mathcal{D}^{(L)}$ is the number of all possible colored Dick paths with $q$ colors, which is

$$\mathcal{D}^{(L)} = q^{\frac{L}{2}} C\left(\frac{L}{2}\right) p_L, \tag{13}$$

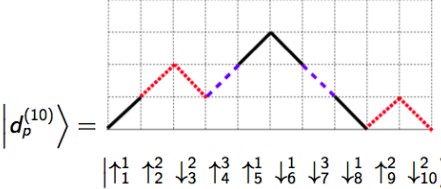
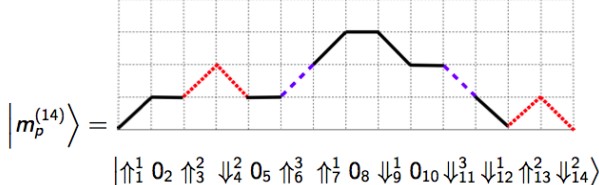

Figure 1: An example of Dyck path (left panel) and of a Motzkin path (right panel) with $q = 3$ colors, which contribute to the ground states for the 3-color spin models, respectively, the spin-$\frac{5}{2}$ Fredkin model and the spin-3 Motzkin model.

where $C(n) = \frac{2n!}{n!(n+1)!}$ are the Catalan numbers and $p_n = (1 - \mathrm{mod}(n, 2))$ selects even integers. Analogously, the ground state of the Motzkin Hamiltonian is a uniform superpositions of all possible Motzkin paths

$$|\mathcal{P}^{(L)}\rangle = \frac{1}{\sqrt{\mathcal{M}^{(L)}}} \sum_p |m_p^{(L)}\rangle, \tag{14}$$

where the normalization factor

$$\mathcal{M}^{(L)} = \sum_{\ell=0}^{\lfloor \frac{L}{2} \rfloor} q^\ell \begin{pmatrix} L \\ 2\ell \end{pmatrix} C(\ell) \tag{15}$$

is the colored Motzkin number, i.e. the number of all the possible colored Motzkin paths. Because of this mapping between the ground states and the lattice paths, several ground state properties can be studied exactly resorting to combinatorics.

## 3.1 Decomposition in two parts

The ground state for both the models can be written in terms of states defined on two subsystems, $A$ and $B$, as it follows

$$|\mathcal{P}^{(L)}\rangle = \sum_{h=0}^{h_m} \sqrt{\mathcal{A}_h} \sum_{c_1, \dots, c_h} |\mathcal{P}_{0h}^{(\ell_A)}\rangle_{c_1, \dots, c_h} |\mathcal{P}_{h0}^{(L-\ell_A)}\rangle_{c_h, \dots, c_1}, \tag{16}$$

where, $h_m = \min(\ell_A, L - \ell_A)$ and $\mathcal{A}_h$ are some Schmidt coefficients depending of the number of paths, whose expressions will be given in the next section for the two cases, in Eq. (19) for the Fredkin model and Eq. (66) for the Motzkin one.
$|\mathcal{P}_{0h}^{(\ell_A)}\rangle_{c_1, \dots, c_h}$ is an orthonormal state defined on the subsystem $A$ made by a uniform superposition of lattice paths (of the Motzkin or Dyck type) which start from the origin and reach the height $h$ after $\ell_A$ steps, with, therefore, $h$ unmatched up-spins with indices $c_1, \dots, c_h$. Analogously, $|\mathcal{P}_{h0}^{(L-\ell_A)}\rangle_{c_1, \dots, c_h}$ is an orthonormal state defined on the subsystem $B$ made by a uniform superposition of lattice paths (of the Motzkin or Dyck type) which start from the point $(\ell_A, h)$ and reach the ending point $(L, 0)$ after $\ell_B = L - \ell_A$ steps, with $h$ unmatched down-spins with indices $c_1, \dots, c_h$.

## 3.2 Decomposition in three parts

Let us now divide our spin chains in three parts, a left and a right part, $A$ and $B$, and a central part $C$, see Fig 2. The ground state can decomposed in terms of states defined in these three regions as it follows

$$|\mathcal{P}^{(L)}\rangle = \sum_{h=0}^{h_m} \sum_{h'=0}^{h'_m} \sum_{z=0}^{\min(h,h')} \sqrt{\mathcal{A}_{hh'z}} \sum_{\substack{c_1, \dots, c_h \\ \bar{c}_1, \dots, \bar{c}_{h'}}} |\mathcal{P}_{0h}^{(\ell_A)}\rangle_{c_1, \dots, c_h} |\mathcal{P}_{hh'(z)}^{(L-\ell_A-\ell_B)}\rangle_{\substack{c_h, \dots, c_1 \\ \bar{c}_1, \dots, \bar{c}_{h'}}} |\mathcal{P}_{h'0}^{(\ell_B)}\rangle_{\bar{c}_{h'}, \dots, \bar{c}_1}, \tag{17}$$

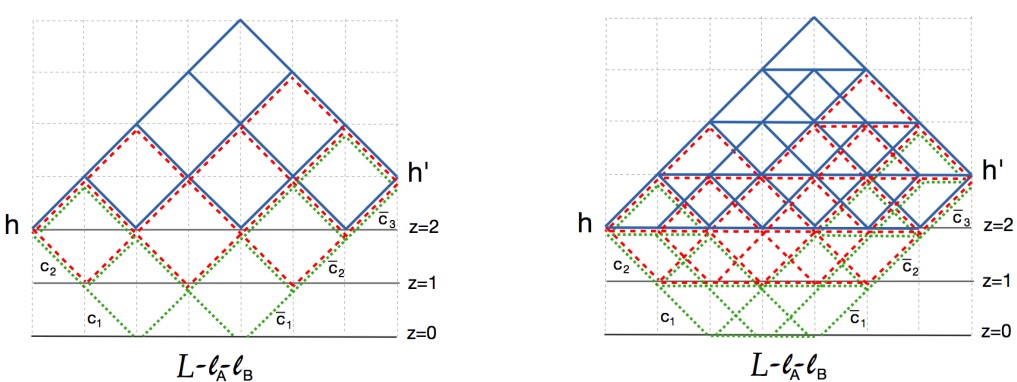

Figure 2: Tripartition of a spin chain into subsystems A, B and C.

with $h_m = \min(\ell_A, L - \ell_A)$, $h'_m = \min(\ell_B, L - \ell_B)$ and where $|\mathcal{P}_{h0}^{(L-\ell_A)}\rangle_{c_1,\dots,c_h}$ an orthonormal state defined on the region $A$ as in Eq. (16), namely as a uniform superposition of lattice paths starting from the origin and ending at $(\ell_A, h)$ with $h$ unmatched colored up-spins and $|\mathcal{P}_{h'0}^{(\ell_B)}\rangle_{\bar{c}_{h'},\dots,\bar{c}_1}$ a uniform superposition of lattice paths defined on $B$, starting from the point $(L - \ell_B, h')$ and ending at $(L, 0)$ with $h'$ unmatched colored down-spins. Moreover

$$|\mathcal{P}_{hh'(z)}^{(L-\ell_A-\ell_B)}\rangle_{\substack{c_h,\dots,c_1 \\ \bar{c}_1,\dots,\bar{c}_{h'}}} = \delta_{c_1 \bar{c}_1} \dots \delta_{c_z \bar{c}_z} |\mathcal{P}_{h-z\,h'-z}^{(L-\ell_A-\ell_B)}\rangle_{\substack{c_h,\dots,c_{z+1} \\ \bar{c}_{z+1},\dots,\bar{c}_{h'}}} \tag{18}$$

is the orthonormal state uniformly composed by all the paths with $(L - \ell_A - \ell_B)$ steps starting at height $h$ and ending at height $h'$, with $(h - z)$ unmatched down-spins and $(h' - z)$ unmatched up-spins, namely those paths which touch at least once the horizontal line defined by $z$ but never cross it. In our notation the indices in Eq. (18) for unmatched spins are useful also for the colorless case to classify the paths by the level $z$. Actually the minimum value of $z$ which contributes to the sum appearing in Eq. (17) is $z_{min} = \max\left(0, \lceil \frac{h + h' - (L - \ell_A - \ell_B)}{2} \rceil \right)$.

This quantity, i.e. the horizon $z$, can be seen, therefore, as a quantum number classifying all the states in the central region $C$, since for any $z$ the states $|\mathcal{P}_{hh'(z)}^{(L-\ell_A-\ell_B)}\rangle_{\substack{c_1,\dots c_h \\ \bar{c}_1,\dots \bar{c}_{h'}}}$ are orthogonal to each other simply because composed by local spin states, expressed in the canonical orthogonal basis. An example of this classification is shown in Fig. 3 for the Fredkin and the Motzkin cases.

Figure 3: Examples of Dyck (left) and Motzkin (right) paths to be taken in a central region $C$ of length $L - \ell_A - \ell_B = 7$ after a tripartition, which start at height $h = 2$ and end at height $h' = 3$, classified by touching at least once, but not crossing, the horizontal lines $z = 0, 1, 2$. The solid blue lines are all the paths which touch $z = 2$, that contribute to $|\mathcal{P}_{01}^{(7)}\rangle_{\bar{c}_3}$, the dashed red lines are all the paths which touch $z = 1$, that contribute to $|\mathcal{P}_{12}^{(7)}\rangle_{c_2, \bar{c}_2, \bar{c}_3}$, the dotted green lines are all those which touch $z = 0$, that contribute to $|\mathcal{P}_{23}^{(7)}\rangle_{c_2, c_1, \bar{c}_1, \bar{c}_2, \bar{c}_3}$.

# 4  Entanglement properties of the Fredkin chain

We will study the entanglement properties of the ground state for the Fredkin model, reviewing the entanglement entropy after a bipartition, and then calculating the negativity and the mutual information shared by the two spins at the edges, resorting to the decompositions in Eqs. (16), (17). We will show that these quantities, particularly the mutual information, revel an unconventional long-distance behavior. Before to proceed we need to know the coefficient in Eq. (16) for the Fredkin ground state, decomposed in two parts, which is the Schmidt number resulting from the product of the normalization factors of $|\mathcal{P}_{0h}^{(\ell_A)}\rangle$ and $|\mathcal{P}_{h0}^{(L-\ell_A)}\rangle$, when expressed in the canonical basis, divided by the normalization factor of $|\mathcal{P}^{(L)}\rangle$ and the number of color degrees of freedom for $h$ up-spins [9,10]

$$\mathcal{A}_h = \frac{\mathcal{D}_{0h}^{(\ell_A)} \mathcal{D}_{h0}^{(L-\ell_A)} q^{-h}}{\mathcal{D}^{(L)}}. \tag{19}$$

The coefficient in Eq. (17), for the decomposition into three parts, analogously to Eq. (19), is given by the product of the normalization factors of $|\mathcal{P}_{0h}^{(\ell_A)}\rangle$, $|\mathcal{P}_{hh'(z)}^{(L-\ell_A-\ell_B)}\rangle$ and $|\mathcal{P}_{h'0}^{(\ell_B)}\rangle$, when these states are expressed in the canonical basis, divided by the normalization factor of $|\mathcal{P}^{(L)}\rangle$ and the total number of color degrees of freedom for $(h+h'-z)$ up-spins, so that it reads

$$\mathcal{A}_{hh'z} = \frac{\mathcal{D}_{0h}^{(\ell_A)} \mathcal{D}_{h'0}^{(\ell_B)} \left( \mathcal{D}_{h-z\,h'-z}^{(L-\ell_A-\ell_B)} - \mathcal{D}_{h-z-1\,h'-z-1}^{(L-\ell_A-\ell_B)} \right) q^{z-h'-h}}{\mathcal{D}^{(L)}}, \tag{20}$$

where

$$\mathcal{D}_{hh'}^{(n)} = q^{\frac{n+h'-h}{2}} \left[ \binom{n}{\frac{n+|h-h'|}{2}} - \binom{n}{\frac{n+h+h'}{2}+1} \right] p_{n+h+h'} \tag{21}$$

is the number of colored Dyck-like paths ($q$ the number of colors) between two points at positive heights $h$ and $h'$ with $n$ steps. We assume $\mathcal{D}_{hh'}^{(n)}$ to be zero for negative $h$ or $h'$ by definition. In particular we have $\mathcal{D}_{00}^{(n)} \equiv \mathcal{D}^{(n)} = q^{\frac{n}{2}} C\left(\frac{n}{2}\right) p_n$. Moreover we notice that $\mathcal{D}_{0h}^{(n)} = q^{\frac{n+h}{2}} \frac{h+1}{\frac{n+h}{2}+1} \binom{n}{\frac{n+h}{2}} p_{n+h} = q^h \mathcal{D}_{h0}^{(n)}$. The quantity in the brackets of Eq. (20) counts the number of just those paths which touch but not cross the level $z$. For example, in Fig. 3, for $z = 1$ the paths are only those depicted by dashed red lines.

## 4.1  Entanglement entropy

In this section we will briefly review the calculation for the von Neumann entanglement entropy [9,10]. The reduced density matrix after a bipartition of the whole systems into two subsystems $A$ and $B$, after tracing out one of them, is obtained from Eq. (16)

$$\rho_A = \mathrm{Tr}_B |\mathcal{P}^{(L)}\rangle \langle \mathcal{P}^{(L)}| = \sum_h^{h_m} \mathcal{A}_h \sum_{c_1,\ldots,c_h} |\mathcal{P}_{0h}^{(\ell_A)}\rangle_{c_1,\ldots,c_h} \langle \mathcal{P}_{0h}^{(\ell_A)}|_{c_1,\ldots,c_h}, \tag{22}$$

where $\mathcal{A}_h$ is given by Eq. (19). Since, for any $h$, there are $q^h$ eigenvalues equal to $\mathcal{A}_h$, the entanglement entropy is simply

$$S_A = -\sum_h^{h_m} q^h \mathcal{A}_h \log(\mathcal{A}_h) = \sum_h^{h_m} \frac{\mathcal{D}_{0h}^{(\ell_A)} \mathcal{D}_{h0}^{(L-\ell_A)}}{\mathcal{D}^{(L)}} \left[ h \log q - \log \left( \frac{\mathcal{D}_{0h}^{(\ell_A)} \mathcal{D}_{h0}^{(L-\ell_A)}}{\mathcal{D}^{(L)}} \right) \right]. \tag{23}$$

Since $q^h \mathcal{A}_h = \frac{\mathcal{D}_{0h}^{(\ell_A)} \mathcal{D}_{h0}^{(L-\ell_A)}}{\mathcal{D}^{(L)}}$ is a normalized probability, $\sum_h q^h \mathcal{A}_h = 1$, the first term of Eq. (23) is $\log q$ times the average height of the paths at a given position located at distance $\ell_A$ from the edge

$$\langle h \rangle_{\ell_A} = \sum_h h \frac{\mathcal{D}_{0h}^{(\ell_A)} \mathcal{D}_{h0}^{(L-\ell_A)}}{\mathcal{D}^{(L)}}, \qquad (24)$$

which, for large $L$ and $\ell_A$, when the binomial factors can be approximated by gaussian factors and the sum by an integral, scales as a square root, $\langle h \rangle_{\ell_A} \simeq \frac{2\sqrt{2}}{\sqrt{\pi}} \sqrt{\frac{\ell_A(L-\ell_A)}{L}}$. The second term, instead scales as $\frac{1}{2} \log\left(\frac{\ell_A(L-\ell_A)}{L}\right)$, therefore, for large systems and for a sizable bipartition one gets

$$S_A \simeq \frac{2\sqrt{2}}{\sqrt{\pi}} \sqrt{\frac{\ell_A(L-\ell_A)}{L}} \log q + \frac{1}{2} \log\left(\frac{\ell_A(L-\ell_A)}{L}\right) + O(1). \qquad (25)$$

Notice that this approximation is very good when the bipartition occurs in the bulk while Eq. (23) is exact for any $\ell_A$ and $L$. For instance, if $\ell_A = 1$, the entanglement entropy, from Eq. (23), is exactly $S_A = \log q$, for any $L$, while Eq. (25) deviates from it.

## 4.2 Reduced density matrix for the edges

Let us consider the system $A \cup B$ made by the two spins located at the edges of our spin chains, as shown in Fig. 4. We will study the entanglement properties between these two spins at the edges for the Fredkin spin chain, tracing out all the spins between the first and the last one

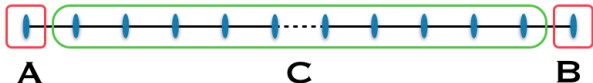

**A**        **C**        **B**

Figure 4: Tripartition of a spin chain in three subsystems $A, B, C$, where the separated regions, $A$ and $B$ are made by single spins at the edges.

described by

$$|\mathcal{P}_{01}^{(\ell_A=1)}\rangle_c = |\uparrow_1^c\rangle, \qquad (26)$$

$$|\mathcal{P}_{10}^{(\ell_B=1)}\rangle_{\bar{c}} = |\downarrow_L^{\bar{c}}\rangle, \qquad (27)$$

so that Eq. (17), dropping the site indices to simplify notation, reads

$$|\mathcal{P}^{(L)}\rangle = \sum_{c,\bar{c}} |\uparrow^c\rangle \left( \sqrt{\mathcal{A}_{110}} |\mathcal{P}_{11}^{(L-2)}\rangle_{c,\bar{c}} + \delta_{c\bar{c}} \sqrt{\mathcal{A}_{111}} |\mathcal{P}^{(L-2)}\rangle \right) |\downarrow^{\bar{c}}\rangle. \qquad (28)$$

The joint reduced density matrix of the subsystem $A \cup B$, after tracing out all the degrees of freedom of the central part, and keeping only the two spins at the edges, is

$$\rho_{AB} = \text{Tr}_C |\mathcal{P}^{(L)}\rangle \langle \mathcal{P}^{(L)}| = \sum_{c,\bar{c}} \left( \mathcal{A}_{111} |\uparrow^c\rangle |\downarrow^{\bar{c}}\rangle \langle\uparrow^{\bar{c}}| \langle\downarrow^{\bar{c}}| + \mathcal{A}_{110} |\uparrow^c\rangle |\downarrow^{\bar{c}}\rangle \langle\uparrow^c| \langle\downarrow^{\bar{c}}| \right), \qquad (29)$$

where the coefficients, from Eq. (20), are

$$\mathcal{A}_{111} = \frac{\mathcal{D}^{(L-2)}}{\mathcal{D}^{(L)}}, \qquad (30)$$

$$\mathcal{A}_{110} = \frac{1}{q} \left( \frac{\mathcal{D}_{11}^{(L-2)} - \mathcal{D}^{(L-2)}}{\mathcal{D}^{(L)}} \right) = \frac{1}{q} \left( \frac{1}{q} - \frac{\mathcal{D}^{(L-2)}}{\mathcal{D}^{(L)}} \right). \qquad (31)$$

The normalization condition is fulfilled since the trace of $\rho_{AB}$ is

$$\mathrm{Tr}\rho_{AB} = q\,\mathcal{A}_{111} + q^2\mathcal{A}_{110} = 1. \tag{32}$$

On the basis $\left(|\uparrow^1\rangle\,|\downarrow^1\rangle\,,|\uparrow^2\rangle\,|\downarrow^2\rangle\,,\dots,|\uparrow^q\rangle\,|\downarrow^q\rangle\,,|\uparrow^1\rangle\,|\downarrow^2\rangle\,,|\uparrow^2\rangle\,|\downarrow^1\rangle\,,\dots,|\uparrow^1\rangle\,|\downarrow^q\rangle\,,|\uparrow^q\rangle\,|\downarrow^1\rangle\,,\right.$ $\left.|\uparrow^2\rangle\,|\downarrow^3\rangle\,,|\uparrow^3\rangle\,|\downarrow^2\rangle\,,\dots\right)^t$, where $t$ means transpose, we can write the $q^2 \times q^2$ reduced density matrix as it follows

$$\rho_{AB} = \begin{pmatrix} \mathcal{A}_{111}\,\mathbb{J}_{q\times q} + \mathcal{A}_{110}\,\mathbb{1}_{q\times q} & \mathbb{0}_{q\times(q^2-q)} \\ \mathbb{0}_{(q^2-q)\times q} & \mathcal{A}_{110}\,\mathbb{1}_{(q^2-q)\times(q^2-q)} \end{pmatrix}, \tag{33}$$

where $\mathbb{J}$ is a matrix with all elements equal to 1, $\mathbb{0}$ a matrix with all zeros and $\mathbb{1}$ the identity matrix.

## 4.3 Negativity

We calculate now the quantum negativity which detects the entanglement between two disjoint regions and can be defined as it follows

$$\mathcal{N} = \frac{1}{2}\sum_\alpha (|\lambda_\alpha| - \lambda_\alpha), \tag{34}$$

where $\lambda_\alpha$ are the eigenvalues of the partial transpose of the reduced density matrix with respect to a region ($B$, for instance), obtained by transposing the indices related to the degrees of freedom of one part. The partial transpose of $\rho_{AB}$ with respect to $B$, from Eq. (29), is

$$\rho_{AB}^{t_B} = \sum_{c,\bar{c}} \left( \mathcal{A}_{111}\,|\uparrow^c\rangle\,|\downarrow^{\bar{c}}\rangle\,\langle\uparrow^{\bar{c}}|\,\langle\downarrow^c| + \mathcal{A}_{110}\,|\uparrow^c\rangle\,|\downarrow^{\bar{c}}\rangle\,\langle\uparrow^c|\,\langle\downarrow^{\bar{c}}| \right). \tag{35}$$

Taking the same basis as for Eq. (33), the partial transpose of the reduced density matrix reads

$$\rho_{AB}^{t_B} = \begin{pmatrix} (\mathcal{A}_{111} + \mathcal{A}_{110})\mathbb{1}_{q\times q} & \mathbb{0}_{q\times(q^2-q)} \\ \mathbb{0}_{(q^2-q)\times q} & \mathcal{A}_{110}\,\mathbb{1}_{\frac{(q^2-q)}{2}\times\frac{(q^2-q)}{2}}\otimes\mathbb{A} \end{pmatrix}, \tag{36}$$

where the last block is a Kronecker product of an identity matrix and a $2\times 2$ matrix

$$\mathbb{A} = \begin{pmatrix} \mathcal{A}_{110} & \mathcal{A}_{111} \\ \mathcal{A}_{111} & \mathcal{A}_{110} \end{pmatrix}. \tag{37}$$

The eigenvalues of $\rho_{AB}^{t_B}$ are $(\mathcal{A}_{111} + \mathcal{A}_{110})$ with multiplicity $q(q+1)/2$ and $(\mathcal{A}_{110} - \mathcal{A}_{111})$ with multiplicity $q(q-1)/2$, therefore the negativity is greater than zero if $\mathcal{A}_{111} > \mathcal{A}_{110}$, namely, if

$$q\frac{\mathcal{D}^{(L-2)}}{\mathcal{D}^{(L)}} = \frac{C(L/2-1)}{C(L/2)} = \frac{L+2}{4(L-1)} > \frac{1}{q+1}, \tag{38}$$

which is verified for $q \geq 3$, for any finite $L$. Therefore $\mathcal{N} = 0$ for $q \leq 2$ (and $q = 3$ in the limit $L \to \infty$) while

$$\mathcal{N} = \frac{(q-1)}{2}\left( (q+1)\frac{\mathcal{D}^{(L-2)}}{\mathcal{D}^{(L)}} - \frac{1}{q} \right), \text{ for } q \geq 3. \tag{39}$$

Some values of $\mathcal{N}$ as a function of the color number $q$ are reported in Fig. 5. In the large $L$ limit the negativity does not vanishes for any $q > 3$ and goes to

$$\mathcal{N} \xrightarrow[L\to\infty]{} \frac{(q-1)(q-3)}{8q}. \tag{40}$$

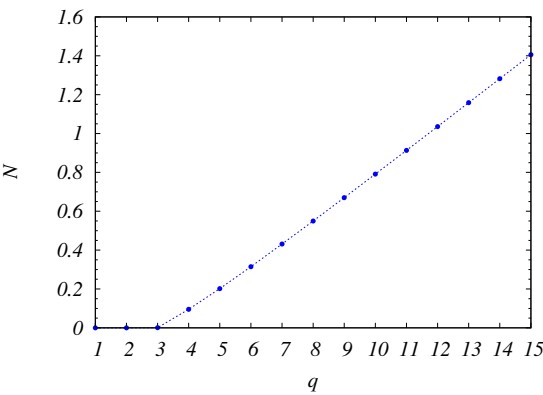

Figure 5: Negativity for two spins at the edges of a Fredkin chain with $L = 1000$, as a function of the number of colors $q$.

We found, therefore, that, for $q > 3$, the negativity $\mathcal{N}$ does not vanish even at infinite distance between the two spins at the edges of the chain. This means that the quantum state in Eq. (29) is surely distantly entangled. On the other hand, for $q = 1$ the state is separable while for $q = 2$ the state described by Eq. (29) is a Werner state with zero negativity, nevertheless it is entangled, is a so-called PPT entangled state, as we will show in the next section using another entanglement measure. For $q = 3$ the negativity is $\mathcal{N} = 1/(L-1)$, therefore the state is surely entangled for any finite $L$. Also in this case we will see that, even if the negativity goes to zero for infinite distance, the state in Eq. (29) is long-distance entangled, as shown by the following calculation.

## 4.4 Mutual Information

The eigenvalues of the reduced density matrix $\rho_{AB}$, from Eq. (33), are $(\mathcal{A}_{110} + q\mathcal{A}_{111})$ and $\mathcal{A}_{110}$, the latter with multiplicity $(q^2 - 1)$, so that the entanglement entropy is

$$S_{AB} = -(q^2-1)\mathcal{A}_{110}\log(\mathcal{A}_{110}) - (\mathcal{A}_{110} + q\mathcal{A}_{111})\log(\mathcal{A}_{110} + q\mathcal{A}_{111}), \qquad (41)$$

with $\mathcal{A}_{111}$ and $\mathcal{A}_{110}$ given by Eqs. (30) and (31). On the other hand, from Eq. (19), since $\mathcal{A}_1 = q^{-1}$ which is the eigenvalue of $\rho_A$ (and $\rho_B$) with multiplicity $q$, we have

$$S_A = S_B = -q\mathcal{A}_1\log(\mathcal{A}_1) = \log q. \qquad (42)$$

From these results we can calculate and study another entanglement measure which is the mutual information

$$\mathcal{I}_{AB} = S_A + S_B - S_{AB} \qquad (43)$$

as a function of the size $L$, being $L-2$ the distance between the two disjoint spins in $A$ and $B$, and also $\mathcal{I}_{AB}$ as a function the color number $q$.

**Colorless case** : For $q = 1$, we have $S_{AB} = 0$ as well as $S_A = S_B = 0$, therefore $\mathcal{I}_{AB} = 0$ exactly, for any size of the chain $L$. This is due to the fact that the first and the last spins of the colorless Fredkin model are uncorrelated in the ground state. For that reason one has to increase the size of the subsystems $A$ and $B$ including further spins, as done in the next section, Sec. 4.5, where we will consider two spins at each edge, revealing in this way that there is a long-distance entanglement even for colorless case.

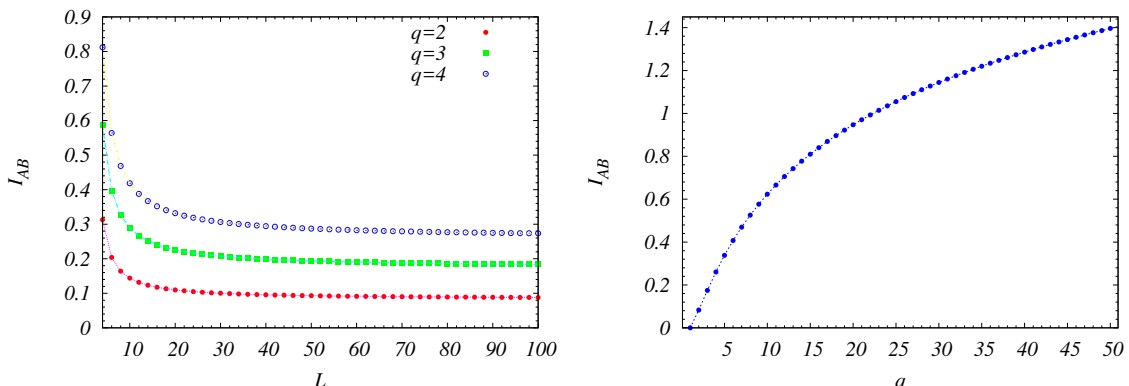

Figure 6: (Left) Mutual information between two spins at the edges of colorful Fredkin chains, as a function of the size $L$, for different values of $q$. (Right) Mutual Information at long distance, $L \to \infty$, between two spins at the edges of colorful Fredkin chains, as a function of the color number $q$.

**Colorful case** : For colorful cases ($q > 1$), instead, $\mathcal{I}_{AB}$ turns to be finite also for large distances, namely for large $L$ ($L \gg 1$), as shown in Fig. 6. Actually we can calculate the limit of $L \to \infty$, since $\mathcal{I}_{AB}$ can be written in terms of $\frac{\mathcal{D}^{(L-2)}}{\mathcal{D}^{(L)}}$ only and

$$\lim_{L \to \infty} \frac{\mathcal{D}^{(L-2)}}{\mathcal{D}^{(L)}} = \frac{1}{4q}, \tag{44}$$

where lim means the limit of a sequence, therefore $\mathcal{A}_{111} \to \frac{1}{4q}$ and $\mathcal{A}_{110} \to \frac{3}{4q^2}$, so that

$$\mathcal{I}_{AB} \underset{L \to \infty}{\longrightarrow} \left[ 2 \log q + \frac{(3+q^2)}{4q^2} \log\left(\frac{3+q^2}{4q^2}\right) + \frac{3(q^2-1)}{4q^2} \log\left(\frac{3}{4q^2}\right) \right]. \tag{45}$$

We show therefore that the two spins located at the edges of the chain, even when the distance is infinite, are strongly entangled for any $q > 1$. This behavior is consistent with the violation of the cluster decomposition occurring in such colorful cases.

## 4.5 Entanglement between the two couples of spins at the edges

As we know, for the colorless case, the first and the last spins are completely uncorrelated in the ground state, since for all the configurations of the spins in the bulk which contribute to the ground state, the first spin is always up and the last is always down. For colorful case, instead, they are correlated because of the color matching condition. For that reason we will consider more than one spin at the edges, studying the entanglement properties of two couples of spins at the borders, namely $A$ and $B$ made by two spins instead of one, as shown in Fig. 7.

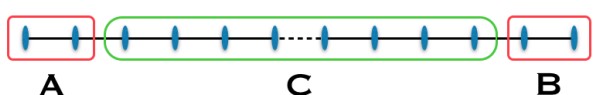

Figure 7: Tripartition of a spin chain in three subsystems, where the separated regions at the edges, $A$ and $B$, are both made by two spins.

In this case, for any $q$ the states at the edges are given by

$$|\mathcal{P}_{02}^{(\ell_A=2)}\rangle_{c_1 c_2} = |\uparrow_1^{c_1} \uparrow_2^{c_2}\rangle, \tag{46}$$

$$|\mathcal{P}_{20}^{(\ell_B=2)}\rangle_{\bar{c}_1 \bar{c}_2} = |\downarrow_{L-1}^{\bar{c}_1} \downarrow_L^{\bar{c}_2}\rangle, \tag{47}$$

$$|\mathcal{P}_{00}^{(\ell_A=2)}\rangle = |\mathcal{P}^{(2)}\rangle = \sum |\uparrow_1^c \downarrow_2^c\rangle \equiv |\uparrow\downarrow\rangle, \tag{48}$$

$$|\mathcal{P}_{00}^{(\ell_B=2)}\rangle = |\mathcal{P}^{(2)}\rangle = \sum_c |\uparrow_{L-1}^c \downarrow_L^c\rangle \equiv |\uparrow\downarrow\rangle, \tag{49}$$

so that, dropping the site indices to simplify the notation, Eq. (17) reads

$$
\begin{aligned}
|\mathcal{P}^{(L)}\rangle &= \sqrt{\mathcal{A}_{000}}\,|\uparrow\downarrow\rangle\,|\mathcal{P}^{(L-4)}\rangle\,|\uparrow\downarrow\rangle + \sqrt{\mathcal{A}_{200}}\sum_{c_1,c_2} |\uparrow^{c_1}\uparrow^{c_2}\rangle\,|\mathcal{P}_{20}^{(L-4)}\rangle_{c_2 c_1}\,|\uparrow\downarrow\rangle \\
&+ \sqrt{\mathcal{A}_{020}}\sum_{\bar{c}_1,\bar{c}_2} |\uparrow\downarrow\rangle\,|\mathcal{P}_{02}^{(L-4)}\rangle_{\bar{c}_1 \bar{c}_2}\,|\downarrow^{\bar{c}_2}\downarrow^{\bar{c}_1}\rangle + \sqrt{\mathcal{A}_{220}}\sum_{\substack{c_1,c_2\\\bar{c}_1,\bar{c}_2}} |\uparrow^{c_1}\uparrow^{c_2}\rangle\,|\mathcal{P}_{22}^{(L-4)}\rangle_{\substack{c_2,c_1\\\bar{c}_1,\bar{c}_2}}\,|\downarrow^{\bar{c}_2}\downarrow^{\bar{c}_1}\rangle \\
&+ \sqrt{\mathcal{A}_{221}}\sum_{c_1,c_2,\bar{c}_2} |\uparrow^{c_1}\uparrow^{c_2}\rangle\,|\mathcal{P}_{11}^{(L-4)}\rangle_{c_2,\bar{c}_2}\,|\downarrow^{\bar{c}_2}\downarrow^{c_1}\rangle + \sqrt{\mathcal{A}_{222}}\sum_{c_1,c_2} |\uparrow^{c_1}\uparrow^{c_2}\rangle\,|\mathcal{P}^{(L-4)}\rangle\,|\downarrow^{c_2}\downarrow^{c_1}\rangle,
\end{aligned}
\tag{50}
$$

where the coefficients are

$$\mathcal{A}_{000} = \frac{(\mathcal{D}^{(2)})^2 \mathcal{D}^{(L-4)}}{\mathcal{D}^{(L)}} = q^2 \frac{\mathcal{D}^{(L-4)}}{\mathcal{D}^{(L)}}, \tag{51}$$

$$\mathcal{A}_{200} = \frac{\mathcal{D}^{(2)}\mathcal{D}_{20}^{(L-4)}}{\mathcal{D}^{(L)}} = q\frac{\mathcal{D}_{20}^{(L-4)}}{\mathcal{D}^{(L)}}, \tag{52}$$

$$\mathcal{A}_{020} = \frac{1}{q^2}\frac{\mathcal{D}^{(2)}\mathcal{D}_{02}^{(L-4)}}{\mathcal{D}^{(L)}} = \frac{1}{q}\frac{\mathcal{D}_{02}^{(L-4)}}{\mathcal{D}^{(L)}} = \mathcal{A}_{200}, \tag{53}$$

$$\mathcal{A}_{220} = \frac{1}{q^2}\left(\frac{\mathcal{D}_{22}^{(L-4)} - \mathcal{D}_{11}^{(L-4)}}{\mathcal{D}^{(L)}}\right), \tag{54}$$

$$\mathcal{A}_{221} = \frac{1}{q}\left(\frac{\mathcal{D}_{11}^{(L-4)} - \mathcal{D}^{(L-4)}}{\mathcal{D}^{(L)}}\right), \tag{55}$$

$$\mathcal{A}_{222} = \frac{(\mathcal{D}_{20}^{(2)})^2 \mathcal{D}^{(L-4)}}{\mathcal{D}^{(L)}} = \frac{\mathcal{D}^{(L-4)}}{\mathcal{D}^{(L)}}. \tag{56}$$

Let us now consider the colorless case ($q = 1$) for simplicity. The reduced density matrix of $A \cup B$, after tracing out over the states of the central region, is

$$
\begin{aligned}
\rho_{AB} &= \mathcal{A}_{000}\,|\uparrow\downarrow\rangle\,|\uparrow\downarrow\rangle\,\langle\uparrow\downarrow|\,\langle\uparrow\downarrow| + \mathcal{A}_{200}\,|\uparrow\uparrow\rangle\,|\uparrow\downarrow\rangle\,\langle\uparrow\uparrow|\,\langle\uparrow\downarrow| + \mathcal{A}_{020}\,|\uparrow\downarrow\rangle\,|\downarrow\downarrow\rangle\,\langle\uparrow\downarrow|\,\langle\downarrow\downarrow| \\
&+ \sqrt{\mathcal{A}_{000}\mathcal{A}_{222}}\left(|\uparrow\downarrow\rangle\,|\uparrow\downarrow\rangle\,\langle\uparrow\uparrow|\,\langle\downarrow\downarrow| + |\uparrow\uparrow\rangle\,|\downarrow\downarrow\rangle\,\langle\uparrow\downarrow|\,\langle\uparrow\downarrow|\right) \\
&+ (\mathcal{A}_{220} + \mathcal{A}_{221} + \mathcal{A}_{222})\,|\uparrow\uparrow\rangle\,|\downarrow\downarrow\rangle\,\langle\uparrow\uparrow|\,\langle\downarrow\downarrow|,
\end{aligned}
\tag{57}
$$

where $(\mathcal{A}_{220} + \mathcal{A}_{221} + \mathcal{A}_{222}) = \frac{\mathcal{D}_{22}^{(L-4)}}{\mathcal{D}^{(L)}}$ and $\mathcal{A}_{000} = \mathcal{A}_{222} = \frac{\mathcal{D}^{(L-4)}}{\mathcal{D}^{(L)}}$.

On the basis $(|\uparrow\uparrow\rangle\,|\uparrow\downarrow\rangle, |\uparrow\downarrow\rangle\,|\uparrow\downarrow\rangle, |\uparrow\uparrow\rangle\,|\downarrow\downarrow\rangle, |\uparrow\downarrow\rangle\,|\downarrow\downarrow\rangle)^t$ the reduced density matrix can be written as

$$
\rho_{AB} = \frac{1}{\mathcal{D}^{(L)}}\begin{pmatrix} \mathcal{D}_{20}^{(L-4)} & 0 & 0 & 0 \\ 0 & \mathcal{D}^{(L-4)} & \mathcal{D}^{(L-4)} & 0 \\ 0 & \mathcal{D}^{(L-4)} & \mathcal{D}_{22}^{(L-4)} & 0 \\ 0 & 0 & 0 & \mathcal{D}_{02}^{(L-4)} \end{pmatrix}, \tag{58}
$$

with $\mathcal{D}_{20}^{(L-4)} = \mathcal{D}_{02}^{(L-4)}$. The corresponding partial transpose matrix with respect to $B$ is

$$
\begin{aligned}
\rho_{AB}^{t_B} = {}& \mathcal{A}_{000} \left|\uparrow\downarrow\right\rangle \left|\uparrow\downarrow\right\rangle \left\langle\uparrow\downarrow\right| \left\langle\uparrow\downarrow\right| + \mathcal{A}_{200} \left|\uparrow\uparrow\right\rangle \left|\uparrow\downarrow\right\rangle \left\langle\uparrow\uparrow\right| \left\langle\uparrow\downarrow\right| + \mathcal{A}_{020} \left|\uparrow\downarrow\right\rangle \left|\downarrow\downarrow\right\rangle \left\langle\uparrow\downarrow\right| \left\langle\downarrow\downarrow\right| \\
& + \sqrt{\mathcal{A}_{000}\mathcal{A}_{222}} \left( \left|\uparrow\downarrow\right\rangle \left|\downarrow\downarrow\right\rangle \left\langle\uparrow\uparrow\right| \left\langle\uparrow\downarrow\right| + \left|\uparrow\uparrow\right\rangle \left|\uparrow\downarrow\right\rangle \left\langle\uparrow\downarrow\right| \left\langle\downarrow\downarrow\right| \right) \\
& + (\mathcal{A}_{220} + \mathcal{A}_{221} + \mathcal{A}_{222}) \left|\uparrow\uparrow\right\rangle \left|\downarrow\downarrow\right\rangle \left\langle\uparrow\uparrow\right| \left\langle\downarrow\downarrow\right|,
\end{aligned}
\tag{59}
$$

which, on the same basis of Eq. (58), reads

$$
\rho_{AB}^{t_B} = \frac{1}{\mathcal{D}^{(L)}} \begin{pmatrix} \mathcal{D}_{20}^{(L-4)} & 0 & 0 & \mathcal{D}^{(L-4)} \\ 0 & \mathcal{D}^{(L-4)} & 0 & 0 \\ 0 & 0 & \mathcal{D}_{22}^{(L-4)} & 0 \\ \mathcal{D}^{(L-4)} & 0 & 0 & \mathcal{D}_{02}^{(L-4)} \end{pmatrix},
\tag{60}
$$

whose eigenvalues are $\frac{\mathcal{D}_{22}^{(L-4)}}{\mathcal{D}^{(L)}}$, $\frac{\mathcal{D}^{(L-4)}}{\mathcal{D}^{(L)}}$ and $\frac{\mathcal{D}_{20}^{(L-4)} \pm \mathcal{D}^{(L-4)}}{\mathcal{D}^{(L)}}$, and since $\mathcal{D}_{20}^{(2n)} \geq \mathcal{D}^{(2n)}$, $\forall n \geq 1$, the negativity is zero, $\mathcal{N} = 0$.

Tracing out the degrees of freedom of one of the two parts we get $\rho_A = \mathrm{Tr}_B \rho_{AB}$, or $\rho_B = \mathrm{Tr}_A \rho_{AB}$ which can be written as

$$
\rho_A = \frac{1}{\mathcal{D}^{(L)}} \begin{pmatrix} \mathcal{D}_{20}^{(L-4)} + \mathcal{D}_{22}^{(L-4)} & 0 \\ 0 & \mathcal{D}^{(L-4)} + \mathcal{D}_{02}^{(L-4)} \end{pmatrix},
$$

$$
\rho_B = \frac{1}{\mathcal{D}^{(L)}} \begin{pmatrix} \mathcal{D}^{(L-4)} + \mathcal{D}_{20}^{(L-4)} & 0 \\ 0 & \mathcal{D}_{02}^{(L-4)} + \mathcal{D}_{22}^{(L-4)} \end{pmatrix}.
\tag{61}
$$

One can notice that $\mathcal{D}_{20}^{(L-4)} + \mathcal{D}_{22}^{(L-4)} = \mathcal{D}_{02}^{(L-2)}$ and $\mathcal{D}^{(L-4)} + \mathcal{D}_{02}^{(L-4)} = \mathcal{D}^{(L-2)}$, so that

$$
\rho_A = \frac{1}{\mathcal{D}^{(L)}} \begin{pmatrix} \mathcal{D}_{02}^{(L-2)} & 0 \\ 0 & \mathcal{D}^{(L-2)} \end{pmatrix}, \quad \rho_B = \frac{1}{\mathcal{D}^{(L)}} \begin{pmatrix} \mathcal{D}^{(L-2)} & 0 \\ 0 & \mathcal{D}_{02}^{(L-2)} \end{pmatrix},
\tag{62}
$$

which are the same matrices we get after a bipartition of the system, from Eqs. (19), (22), with $\ell_A = 2$ (or $\ell_B = 2$), since $\mathcal{D}_{02}^{(2)} = \mathcal{D}^{(2)} = 1$. In the limit $L \to \infty$ the reduced density matrices in Eqs. (60), (62) become

$$
\rho_{AB} \xrightarrow[L \to \infty]{} \frac{1}{16} \begin{pmatrix} 3 & 0 & 0 & 0 \\ 0 & 1 & 1 & 0 \\ 0 & 1 & 9 & 0 \\ 0 & 0 & 0 & 3 \end{pmatrix},
\tag{63}
$$

whose eigenvalues are twice $3/16$ and $\left(5 \pm \sqrt{17}\right)/16$, and

$$
\rho_A \xrightarrow[L \to \infty]{} \frac{1}{4} \begin{pmatrix} 3 & 0 \\ 0 & 1 \end{pmatrix}, \quad \rho_B \xrightarrow[L \to \infty]{} \frac{1}{4} \begin{pmatrix} 1 & 0 \\ 0 & 3 \end{pmatrix}.
\tag{64}
$$

Now one can easily calculate $S_{AB}$, $S_A$ and $S_B$, finding as a result the large $L$ limit for the mutual information

$$
\mathcal{I}_{AB} \xrightarrow[L \to \infty]{} \frac{1}{16} \left[ 2\sqrt{17} \, \mathrm{arcosh}\left( \frac{5}{\sqrt{17}} \right) - 18 \log 3 + 15 \log 2 \right],
\tag{65}
$$

which is $\mathcal{I}_{AB} \approx 0.01745$ using the natural logarithm. We have shown, therefore, that, even for the colorless case, with the lowest value for the entanglement entropy, the mutual information, shared by two couples of spins at the edges, does not go to zero but remain finite also for infinite distance.

# 5 Entanglement properties of the Motzkin chain

As done for the Fredkin model, we will study the entanglement properties of the Motzkin model in the ground state, briefly reviewing the entanglement entropy after a bipartition, then calculating the negativity and the mutual information shared by the two spins at the edges. We will show that also in this case the latter quantity reveals long-distance entanglement.

The coefficients in Eq. (16) for the ground state of the Motzkin chain, after a bipartition, analogously to Eq. (19), are given by [7,8]

$$\mathcal{A}_h = \frac{\mathcal{M}_{0h}^{(\ell_A)} \mathcal{M}_{h0}^{(L-\ell_A)} q^{-h}}{\mathcal{M}^{(L)}}, \tag{66}$$

while the coefficients in Eq. (17), after decomposing the state into three parts, analogously to what found for the Fredkin model in Eq. (20), are given by

$$\mathcal{A}_{hh'z} = \frac{\mathcal{M}_{0h}^{(\ell_A)} \mathcal{M}_{h'0}^{(\ell_B)} \left( \mathcal{M}_{h-z\,h'-z}^{(L-\ell_A-\ell_B)} - \mathcal{M}_{h-z-1\,h'-z-1}^{(L-\ell_A-\ell_B)} \right) q^{z-h'-h}}{\mathcal{M}^{(L)}}, \tag{67}$$

where

$$\mathcal{M}_{hh'}^{(n)} = \sum_{\ell=0}^{\lfloor \frac{n-|h'-h|}{2} \rfloor} \binom{n}{2\ell + |h'-h|} \mathcal{D}_{hh'}^{(2\ell+|h'-h|)} \tag{68}$$

is the number of colored Motzkin-like paths between two points at heights $h$ and $h'$. Moreover $\mathcal{M}_{0h}^{(n)} = q^h \mathcal{M}_{h0}^{(n)}$, and $\mathcal{M}_{00}^{(n)} \equiv \mathcal{M}^{(n)} = \sum_{\ell=0}^{\lfloor \frac{n}{2} \rfloor} q^\ell \binom{n}{2\ell} C(\ell)$ which is the colored Motzkin number.

## 5.1 Entanglement entropy

In this section we briefly review the calculation for the von Neumann entanglement entropy for the Motzkin chain [7–9]. The reduced density matrix after a bipartition of the system into two subsystems $A$ and $B$, after tracing out the degrees of freedom of one of them, is obtained from Eq. (16) and has the same form reported in Eq. (22) where now $\mathcal{A}_h$ is given by Eq. (66). Since $\rho_A$ have $q^h$ eigenvalues equal to $\mathcal{A}_h$, the entanglement entropy reads as it follows

$$S_A = -\sum_h^{h_m} q^h \mathcal{A}_h \log(\mathcal{A}_h) = \sum_h^{h_m} \frac{\mathcal{M}_{0h}^{(\ell_A)} \mathcal{M}_{h0}^{(L-\ell_A)}}{\mathcal{M}^{(L)}} \left[ h \log q - \log \left( \frac{\mathcal{M}_{0h}^{(\ell_A)} \mathcal{M}_{h0}^{(L-\ell_A)}}{\mathcal{M}^{(L)}} \right) \right]. \tag{69}$$

Also in this case, since $q^h \mathcal{A}_h = \frac{\mathcal{M}_{0h}^{(\ell_A)} \mathcal{M}_{h0}^{(L-\ell_A)}}{\mathcal{M}^{(L)}}$ is a normalized probability, $\sum_h q^h \mathcal{A}_h = 1$, the first term of Eq. (69) is $\log q$ times the average height of the paths at a given position located at distance $\ell_A$ from the edge

$$\langle h \rangle_{\ell_A} = \sum_h h \frac{\mathcal{M}_{0h}^{(\ell_A)} \mathcal{M}_{h0}^{(L-\ell_A)}}{\mathcal{M}^{(L)}}, \tag{70}$$

which, for large $L$ and $\ell_A$, scales as square root, $\langle h \rangle_{\ell_A} \simeq \frac{2\sqrt{2\sigma}}{\sqrt{\pi}} \sqrt{\frac{\ell_A(L-\ell_A)}{L}}$, with $\sigma = 2\sqrt{q}/(2\sqrt{q}+1)$. The second term, instead, scales as $\frac{1}{2} \log \left( \frac{\ell_A(L-\ell_A)}{L} \right)$, therefore, for large systems and for a sizable bipartition one gets

$$S_A \simeq \frac{2\sqrt{2\sigma}}{\sqrt{\pi}} \sqrt{\frac{\ell_A(L-\ell_A)}{L}} \log q + \frac{1}{2} \log \left( \frac{\ell_A(L-\ell_A)}{L} \right) + O(1). \tag{71}$$

As for the Fredkin case, this approximation is very good when the bipartition occurs in the bulk while Eq. (69) is exact for any $\ell_A$ and $L$.

## 5.2 Reduced density matrix of the edges

Let us consider the system $A \cup B$ made by the two spins located at the edges of our spin chains, as shown in Fig. 4, and study the entanglement properties between these two spins at the edges of a Motzkin spin chain by tracing out all the spins between the first and the last one described by

$$|\mathcal{P}_{01}^{(\ell_A=1)}\rangle_c = |\Uparrow_1^c\rangle, \tag{72}$$

$$|\mathcal{P}_{10}^{(\ell_B=1)}\rangle_{\bar{c}} = |\Downarrow_L^{\bar{c}}\rangle, \tag{73}$$

$$|\mathcal{P}_{00}^{(\ell_A=1)}\rangle = |\mathcal{P}^{(1)}\rangle = |0_1\rangle, \tag{74}$$

$$|\mathcal{P}_{00}^{(\ell_B=1)}\rangle = |\mathcal{P}^{(1)}\rangle = |0_L\rangle, \tag{75}$$

so that Eq. (17), dropping the site indices to simplify the notation, reads

$$|\mathcal{P}^{(L)}\rangle = \sqrt{\mathcal{A}_{000}}\,|0\rangle\,|\mathcal{P}^{(L-2)}\rangle\,|0\rangle + \sqrt{\mathcal{A}_{100}}\sum_c |\Uparrow^c\rangle\,|\mathcal{P}_{10}^{(L-2)}\rangle_c\,|0\rangle + \sqrt{\mathcal{A}_{010}}\sum_{\bar{c}} |0\rangle\,|\mathcal{P}_{01}^{(L-2)}\rangle_{\bar{c}}\,|\Downarrow^{\bar{c}}\rangle$$

$$+ \sqrt{\mathcal{A}_{110}}\sum_{c,\bar{c}} |\Uparrow^c\rangle\,|\mathcal{P}_{11}^{(L-2)}\rangle_{c,\bar{c}}\,|\Downarrow^{\bar{c}}\rangle + \sqrt{\mathcal{A}_{111}}\sum_c |\Uparrow^c\rangle\,|\mathcal{P}^{(L-2)}\rangle\,|\Downarrow^c\rangle. \tag{76}$$

The joint reduced density matrix of $A \cup B$, after tracing out all the degrees of freedom of the central part $C$ (see Fig. 4) and keeping only the two spins at the edges, is

$$\rho_{AB} = \mathcal{A}_{000}\,|0\rangle\,|0\rangle\,\langle 0|\,\langle 0| + \mathcal{A}_{100}\sum_c |\Uparrow^c\rangle\,|0\rangle\,\langle\Uparrow^c|\,\langle 0| + \mathcal{A}_{010}\sum_c |0\rangle\,|\Downarrow^c\rangle\,\langle 0|\,\langle\Downarrow^c|$$

$$+ \mathcal{A}_{110}\sum_{c,\bar{c}} |\Uparrow^c\rangle\,|\Downarrow^{\bar{c}}\rangle\,\langle\Uparrow^c|\,\langle\Downarrow^{\bar{c}}| + \mathcal{A}_{111}\sum_{c,\bar{c}} |\Uparrow^c\rangle\,|\Downarrow^c\rangle\,\langle\Uparrow^{\bar{c}}|\,\langle\Downarrow^{\bar{c}}|$$

$$+ \sqrt{\mathcal{A}_{000}\mathcal{A}_{111}}\sum_c \big( |0\rangle\,|0\rangle\,\langle\Uparrow^c|\,\langle\Downarrow^c| + |\Uparrow^c\rangle\,|\Downarrow^c\rangle\,\langle 0|\,\langle 0| \big), \tag{77}$$

where the Schmidt coefficients are given by

$$\mathcal{A}_{000} = \frac{\mathcal{M}^{(L-2)}}{\mathcal{M}^{(L)}} = \mathcal{A}_{111}, \tag{78}$$

$$\mathcal{A}_{010} = \frac{1}{q}\frac{\mathcal{M}_{01}^{(L-2)}}{\mathcal{M}^{(L)}}, \tag{79}$$

$$\mathcal{A}_{100} = \frac{\mathcal{M}_{10}^{(L-2)}}{\mathcal{M}^{(L)}} = \mathcal{A}_{010}, \tag{80}$$

$$\mathcal{A}_{110} = \frac{1}{q}\left(\frac{\mathcal{M}_{11}^{(L-2)} - \mathcal{M}^{(L-2)}}{\mathcal{M}^{(L)}}\right). \tag{81}$$

One can verify that the trace of $\rho_{AB}$ is one,

$$\mathrm{Tr}\rho_{AB} = \mathcal{A}_{000} + q\left(\mathcal{A}_{100} + \mathcal{A}_{010} + \mathcal{A}_{111}\right) + q^2\mathcal{A}_{110}$$

$$= \frac{1}{\mathcal{M}^{(L)}}\left(\mathcal{M}^{(L-2)} + \mathcal{M}_{01}^{(L-2)} + q\mathcal{M}_{10}^{(L-2)} + q\mathcal{M}_{11}^{(L-2)}\right) = 1. \tag{82}$$

Writing $\rho_{AB}$ on the basis $\big(|0\rangle\,|0\rangle, |\Uparrow^1\rangle\,|\Downarrow^1\rangle, \ldots, |\Uparrow^q\rangle\,|\Downarrow^q\rangle, |0\rangle\,|\Downarrow^1\rangle, \ldots, |0\rangle\,|\Downarrow^q\rangle, |\Uparrow^1\rangle\,|0\rangle, \ldots,$
$|\Uparrow^q\rangle\,|0\rangle, |\Uparrow^1\rangle\,|\Downarrow^2\rangle, |\Uparrow^2\rangle\,|\Downarrow^1\rangle, \ldots\big)^t$ we get

$$\rho_{AB} = \begin{pmatrix} \mathcal{A}_{000} & \sqrt{\mathcal{A}_{000}\mathcal{A}_{111}}\,\mathbb{J}_{1\times q} & \mathbb{0}_{1\times q} & \mathbb{0}_{1\times q} & \mathbb{0}_{1\times(q^2-q)} \\ \sqrt{\mathcal{A}_{000}\mathcal{A}_{111}}\,\mathbb{J}_{q\times 1} & \mathcal{A}_{111}\mathbb{J}_{q\times q} + \mathcal{A}_{110}\mathbb{1}_{q\times q} & \mathbb{0}_{q\times q} & \mathbb{0}_{q\times q} & \mathbb{0}_{q\times(q^2-q)} \\ \mathbb{0}_{q\times 1} & \mathbb{0}_{q\times q} & \mathcal{A}_{100}\mathbb{1}_{q\times q} & \mathbb{0}_{q\times q} & \mathbb{0}_{q\times(q^2-q)} \\ \mathbb{0}_{q\times 1} & \mathbb{0}_{q\times q} & \mathbb{0}_{q\times q} & \mathcal{A}_{010}\mathbb{1}_{q\times q} & \mathbb{0}_{q\times(q^2-q)} \\ \mathbb{0}_{(q^2-q)\times 1} & \mathbb{0}_{(q^2-q)\times q} & \mathbb{0}_{(q^2-q)\times q} & \mathbb{0}_{(q^2-q)\times q} & \mathcal{A}_{110}\mathbb{1}_{(q^2-q)\times(q^2-q)} \end{pmatrix}. \tag{83}$$

### 5.3 Negativity

We can calculate the negativity defined as in Eq. (34) where now $\lambda_\alpha$ are the eigenvalues of the partial transpose of the reduced density matrix with respect to $B$ in Eq. (77)

$$
\begin{aligned}
\rho_{AB}^{t_B} &= \mathcal{A}_{000} \, |0\rangle \, |0\rangle \, \langle 0| \, \langle 0| + \mathcal{A}_{100} \sum_c |\Uparrow^c\rangle \, |0\rangle \, \langle \Uparrow^c| \, \langle 0| + \mathcal{A}_{010} \sum_c |0\rangle \, |\Downarrow^c\rangle \, \langle 0| \, \langle \Downarrow^c| \\
&\quad + \mathcal{A}_{110} \sum_{c,\bar{c}} |\Uparrow^c\rangle \, |\Downarrow^{\bar{c}}\rangle \, \langle \Uparrow^c| \, \langle \Downarrow^{\bar{c}}| + \mathcal{A}_{111} \sum_{c,\bar{c}} |\Uparrow^c\rangle \, |\Downarrow^{\bar{c}}\rangle \, \langle \Uparrow^{\bar{c}}| \, \langle \Downarrow^c| \\
&\quad + \sqrt{\mathcal{A}_{000} \mathcal{A}_{111}} \sum_c \Big( |0\rangle \, |\Downarrow^c\rangle \, \langle \Uparrow^c| \, \langle 0| + |\Uparrow^c\rangle \, |0\rangle \, \langle 0| \, \langle \Downarrow^c| \Big).
\end{aligned}
\tag{84}
$$

Expressing this matrix on the same basis of Eq. (83) we can write

$$
\rho_{AB}^{t_B} = \begin{pmatrix}
\mathcal{A}_{000} & \mathbb{0}_{1\times q} & \mathbb{0}_{1\times q} & \mathbb{0}_{1\times q} & \mathbb{0}_{1\times(q^2-q)} \\
\mathbb{0}_{q\times 1} & (\mathcal{A}_{111}+\mathcal{A}_{110})\mathbb{1}_{q\times q} & \mathbb{0}_{q\times q} & \mathbb{0}_{q\times q} & \mathbb{0}_{q\times(q^2-q)} \\
\mathbb{0}_{q\times 1} & \mathbb{0}_{q\times q} & \mathcal{A}_{100}\mathbb{1}_{q\times q} & \sqrt{\mathcal{A}_{000}\mathcal{A}_{111}}\,\mathbb{1}_{q\times q} & \mathbb{0}_{q\times(q^2-q)} \\
\mathbb{0}_{q\times 1} & \mathbb{0}_{q\times q} & \sqrt{\mathcal{A}_{000}\mathcal{A}_{111}}\,\mathbb{1}_{q\times q} & \mathcal{A}_{010}\mathbb{1}_{q\times q} & \mathbb{0}_{q\times(q^2-q)} \\
\mathbb{0}_{(q^2-q)\times 1} & \mathbb{0}_{(q^2-q)\times q} & \mathbb{0}_{(q^2-q)\times q} & \mathbb{0}_{(q^2-q)\times q} & \mathbb{1}_{\frac{(q^2-q)}{2}\times\frac{(q^2-q)}{2}} \otimes \mathbb{A}
\end{pmatrix},
\tag{85}
$$

where the last block on the bottom-right corner is a Kronecker product of an identity matrix and a $2 \times 2$ matrix

$$
\mathbb{A} = \begin{pmatrix} \mathcal{A}_{110} & \mathcal{A}_{111} \\ \mathcal{A}_{111} & \mathcal{A}_{110} \end{pmatrix},
\tag{86}
$$

where, according to Eqs. (78-80), $\mathcal{A}_{111} = \mathcal{A}_{000}$ and $\mathcal{A}_{100} = \mathcal{A}_{010}$. The eigenvalues of $\rho_{AB}^{t_B}$ are $\mathcal{A}_{000}$ with multiplicity one, $(\mathcal{A}_{000}+\mathcal{A}_{110})$ with multiplicity $q$, $(\mathcal{A}_{100}+\mathcal{A}_{000})$ with multiplicity $q$, $(\mathcal{A}_{100}-\mathcal{A}_{000})$ with multiplicity $q$, $(\mathcal{A}_{110}+\mathcal{A}_{000})$ with multiplicity $q(q-1)/2$ and $(\mathcal{A}_{110}-\mathcal{A}_{000})$ with multiplicity $q(q-1)/2$.

Since $\mathcal{A}_{100} \geq \mathcal{A}_{000}$, namely $\mathcal{M}_{10}^n \geq \mathcal{M}^n$, $\forall n \geq 1$, the only possibility for having a negativity greater than zero is when $\mathcal{A}_{000} > \mathcal{A}_{110}$, which occurs for

$$
q > \left( \frac{\mathcal{M}_{11}^{(L-2)}}{\mathcal{M}^{(L-2)}} - 1 \right) \underset{L\to\infty}{\longrightarrow} 3.
\tag{87}
$$

Notice that even if $\mathcal{M}_{11}^{(L-2)}$ and $\mathcal{M}^{(L-2)}$ depend on $q$ (see. Eqs. (21), (68), (15)) their ratio, in the limit $L \to \infty$, goes to 4 from below for any $q$. As a result, for any finite $L$, we have

$$
\mathcal{N} = \frac{(q-1)}{2} \left( (q+1)\frac{\mathcal{M}^{(L-2)}}{\mathcal{M}^{(L)}} - \frac{\mathcal{M}_{11}^{(L-2)}}{\mathcal{M}^{(L)}} \right), \text{ for } q \geq 3
\tag{88}
$$

and $\mathcal{N} = 0$ otherwise ($q = 1, 2$), as in the case of the Fredkin model, see Fig. 8. For $L \gg 1$, we have that $\mathcal{M}_{11}^{(L-2)} \to 4\mathcal{M}^{(L-2)}$ for any $q$ so that Eq. (88) becomes

$$
\mathcal{N} \underset{L\gg 1}{\longrightarrow} \frac{(q-1)(q-3)}{2} \frac{\mathcal{M}^{(L-2)}}{\mathcal{M}^{(L)}}.
\tag{89}
$$

Also for the Motzkin model we have that, for $q > 3$, the negativity $\mathcal{N}$ does not vanish even at infinite distance between the two spins at the edges of the chain. This means that the quantum state in Eq. (77) is surely distantly entangled. On the other hand, for $q = 1$ and $q = 2$ the negativity is zero, nevertheless Eq. (77) is a PPT entangled state, as we will show in the

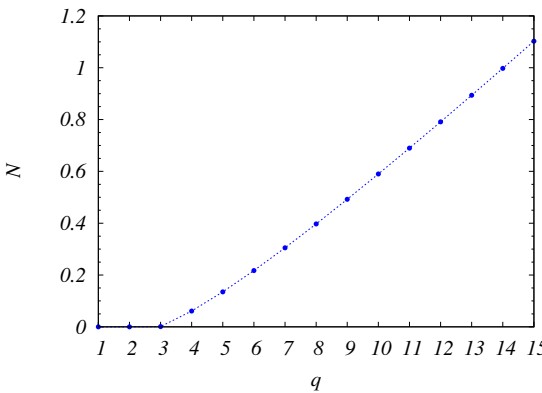

Figure 8: Negativity for two spins at the edges of a Motzkin chain of size $L = 1000$, as a function of the number of colors $q$.

next section. For $q = 3$ the negativity is $\mathcal{N} \approx 7/9L$, therefore the state is surely entangled for any finite $L$, but also in this case we will see that, even if the negativity goes to zero for infinite distance, the state in Eq. (77) is infinitely long-distance entangled, as shown by the following calculation.

## 5.4 Mutual Information

The eigenvalues of the reduced density matrix, Eq. (83), are

$$\lambda_{\pm} = \frac{1}{2}\Big[\mathcal{A}_{110} + (q+1)\mathcal{A}_{000} \pm \sqrt{(\mathcal{A}_{110})^2 + 2(q-1)\mathcal{A}_{000}\mathcal{A}_{110} + (q+1)^2(\mathcal{A}_{000})^2}\Big], \quad (90)$$

together with $\mathcal{A}_{000}$ with multiplicity $(q-1)$, $\mathcal{A}_{100}$ with multiplicity $2q$ and $\mathcal{A}_{110}$ with multiplicity $(q^2 - q)$. The entanglement entropy $S_{AB}$ is, therefore,

$$S_{AB} = -\Big[\lambda_+ \log(\lambda_+) + \lambda_- \log(\lambda_-) + (q-1)\mathcal{A}_{000}\log(\mathcal{A}_{000}) +$$
$$2q\,\mathcal{A}_{100}\log(\mathcal{A}_{100}) + (q^2 - q)\mathcal{A}_{110}\log(\mathcal{A}_{110})\Big], \quad (91)$$

with $\mathcal{A}_{000}$ as in Eq. (78), $\mathcal{A}_{100}$ in Eq. (80) and $\mathcal{A}_{110}$ in Eq. (81). On the other hand

$$S_A = S_B = -\Big[\mathcal{A}_0 \log(\mathcal{A}_0) + q\,\mathcal{A}_1 \log(\mathcal{A}_1)\Big], \quad (92)$$

with the coefficients

$$\mathcal{A}_0 = \frac{\mathcal{M}^{(L-1)}}{\mathcal{M}^{(L)}}, \quad \mathcal{A}_1 = \frac{\mathcal{M}_{10}^{(L-1)}}{\mathcal{M}^{(L)}} \quad (93)$$

fulfilling $\mathcal{A}_0 + q\mathcal{A}_1 = 1$. The mutual information that we report here for convenience

$$\mathcal{I}_{AB} = S_A + S_B - S_{AB} \quad (94)$$

can be calculated for any value of $L$ and $q$ through Eqs. (90), (91) and (92). $\mathcal{I}_{AB}$ as a function of the size $L$ is plotted in Fig. 9 (left panel) for some values of $q$. As one can see, increasing $L$ the mutual information goes to some asymptotic value which depends on $q$. The asymptotic values of $\mathcal{I}_{AB}$ have been calculated and show in Fig. 9 (right panel) for several values of $q$.

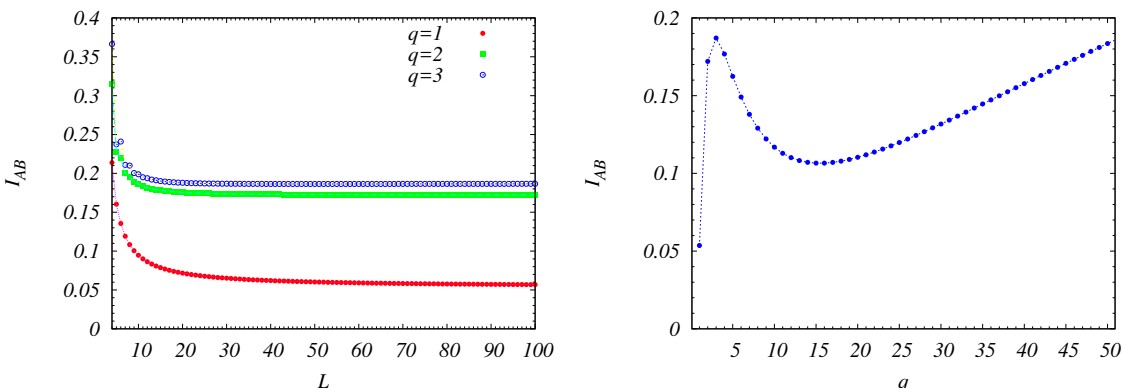

Figure 9: (Left) Mutual information between two spins at the edges of colorful Motzkin chains, as a function of the size $L$, for different values of $q$. (Right) Mutual information at long distance, $L \to \infty$ between two spins at the edges of colorful Motzkin chains, as a function of the color number $q$.

**Colorless case** : Let us focus on the simplest case $q = 1$, in the long distance limit $L \to \infty$. The reduced density matrix, Eq. (83), becomes simply

$$\rho_{AB} \xrightarrow[L \to \infty]{} \frac{1}{9} \begin{pmatrix} 1 & 1 & 0 & 0 \\ 1 & 4 & 0 & 0 \\ 0 & 0 & 2 & 0 \\ 0 & 0 & 0 & 2 \end{pmatrix}, \tag{95}$$

whose eigenvalues are twice $2/9$ and $\left(5 \pm \sqrt{13}\right)/18$. Summing over the right or left spin degrees of freedom we get

$$\rho_A = \rho_B \xrightarrow[L \to \infty]{} \frac{1}{3} \begin{pmatrix} 1 & 0 \\ 0 & 2 \end{pmatrix}, \tag{96}$$

so that the mutual information is given by

$$\mathcal{I}_{AB} \xrightarrow[L \to \infty]{} \frac{1}{18} \left[ 2\sqrt{13} \operatorname{arcosh}\left(\frac{5}{\sqrt{13}}\right) - 16 \log 2 + 5 \log 3 \right], \tag{97}$$

which is $\mathcal{I}_{AB} \approx 0.05361$, in natural logarithm, as shown in Fig. 9.

**Colorful case** : Here we derive an explicit expression for the asymptotic value of $\mathcal{I}_{AB}$ as a function of $q$. For a generic $q$ we can simplify the expression for $\mathcal{I}_{AB}$, in the limit $L \to \infty$, writing it in terms of only one colored Motzkin ratio

$$\mathcal{F}_q \equiv \lim_{L \to \infty} \mathcal{A}_0 = \lim_{L \to \infty} \frac{\mathcal{M}^{(L-1)}}{\mathcal{M}^{(L)}} = \lim_{L \to \infty} \frac{{}_2F_1\left(\frac{1-L}{2}, \frac{2-L}{2}, 2, 4q\right)}{{}_2F_1\left(\frac{-L}{2}, \frac{1-L}{2}, 2, 4q\right)}, \tag{98}$$

since $\mathcal{A}_1 = (1 - \mathcal{A}_0)/q$ and, from Eqs. (78)-(81),

$$\lim_{L \to \infty} \mathcal{A}_{000} = \mathcal{F}_q^2, \tag{99}$$

$$\lim_{L \to \infty} \mathcal{A}_{100} = \frac{2}{\sqrt{q}} \mathcal{F}_q^2, \tag{100}$$

$$\lim_{L \to \infty} \mathcal{A}_{110} = \frac{3}{q} \mathcal{F}_q^2, \tag{101}$$

where we recognize that $\mathcal{M}^{(L)}$ in Eq. (15) can be seen as $_2F_1(\frac{-L}{2}, \frac{1-L}{2}, 2, 4q)$, an hypergeometric serie reduced to a polynomial since at least one on the first two arguments is a nonpositive integer number. Substituting these values in Eq. (90) we get

$$\Lambda_q^\pm \equiv \lim_{L\to\infty} \lambda_\pm = \frac{\mathcal{F}_q^2}{2q}\left(q^2 + q + 3 \pm \sqrt{q^4 + 2q^3 + 7q^2 - 6q + 9}\right) \tag{102}$$

and, from Eqs. (91), (92) we get, for the mutual information, the following asymptotic exact expression

$$\mathcal{I}_{AB} \xrightarrow[L\to\infty]{} \left[2(\mathcal{F}_q - 1)\log\left(\frac{1 - \mathcal{F}_q}{q}\right) - 2\mathcal{F}_q\log\mathcal{F}_q + 2\mathcal{F}_q^2(q-1)\log\mathcal{F}_q\right.$$
$$\left. + 3\mathcal{F}_q^2(q-1)\log\left(\frac{3\mathcal{F}_q^2}{q}\right) + 4\mathcal{F}_q^2\sqrt{q}\log\left(\frac{2\mathcal{F}_q^2}{\sqrt{q}}\right) + \Lambda_q^+\log\Lambda_q^+ + \Lambda_q^-\log\Lambda_q^-\right]. \tag{103}$$

We showed, therefore, that the mutual information shared by the two disjoint spins at the edges is always finite, even when the separation is sent to infinity, also for the colorless case which, from the point of view of the entanglement entropy, resembles a critical system. The expression in Eq. (103), as a function of $q$, perfectly agrees with the results reported in Fig. 9 (right panel) obtained calculating the asymptotic $\mathcal{I}_{AB}$ for several values of $q$. As one can see from Fig. 9, comparing it with Fig. 6, unlike the Fredkin model, because of the greater complexity due to the spin degrees of freedom, $\mathcal{I}_{AB}$ for the Motzkin model does not monotonically increase with $q$.

# 6 Entanglement properties in the bulk

Let us generalize what seen so far considering two disjoint subsystems also in the bulk. This requires to divide our system in five subsystems with different lengths such that $L = \ell_A + \ell_B + \ell_C + \ell_D + \ell_E$, as shown in Fig. 10, and the ground state has to be decomposed in five parts

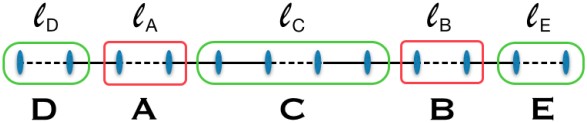

Figure 10: Pentapartition of a spin chain into subsystems A, B, C, D, E.

$$|\mathcal{P}^{(L)}\rangle = \sum_{\substack{h_1 h_2 h_3 h_4 \\ z_1 z_2 z_3}} \sqrt{\mathcal{A}_{\substack{h_1 h_2 h_3 h_4 \\ z_1 z_2 z_3}}} \sum_{\substack{c_1 \dots c_{h_1} \\ \tilde{c}_1 \dots \tilde{c}_{h_2} \\ \hat{c}_1 \dots \hat{c}_{h_4}}} \sum_{\tilde{c}_1 \dots \tilde{c}_{h_3}} |\mathcal{P}^{(\ell_D)}_{0h_1}\rangle_{c_1,\dots,c_{h_1}} |\mathcal{P}^{(\ell_A)}_{h_1 h_2(z_1)}\rangle^{c_{h_1},\dots,c_1}_{\tilde{c}_1,\dots,\tilde{c}_{h_2}} |\mathcal{P}^{(\ell_C)}_{h_2 h_3(z_2)}\rangle^{\tilde{c}_{h_2},\dots,\tilde{c}_1}_{\tilde{c}_1,\dots,\tilde{c}_{h_3}}$$
$$|\mathcal{P}^{(\ell_B)}_{h_3 h_4(z_3)}\rangle^{\tilde{c}_{h_3},\dots,\tilde{c}_1}_{\hat{c}_1,\dots,\hat{c}_{h_4}} |\mathcal{P}^{(\ell_E)}_{h_4 0}\rangle_{\hat{c}_{h_4},\dots,\hat{c}_1}, \tag{104}$$

where the states located in each region is defined as before, see Eq. (18), and the coefficients depend on the model.

## 6.1 Fredkin model

For the half-integer spin model the coefficients appearing in Eq. (104) are the following

$$\mathcal{A}_{\substack{h_1 h_2 h_3 h_4 \\ z_1 z_2 z_3}} = \frac{\mathcal{D}^{(\ell_D)}_{0h_1} \mathcal{L}^{(\ell_A)}_{h_1 h_2 z_1} \mathcal{L}^{(\ell_C)}_{h_2 h_3 z_2} \mathcal{L}^{(\ell_B)}_{h_3 h_4 z_3} \mathcal{D}^{(\ell_E)}_{h_4 0}}{\mathcal{D}^{(L)}} q^{z_1 + z_2 + z_3 - (h_1 + h_2 + h_3 + h_4)}, \tag{105}$$

where

$$\mathcal{L}^{(\ell)}_{h_i h_j z} = \left( \mathcal{D}^{(\ell)}_{h_i - z\, h_j - z} - \mathcal{D}^{(\ell)}_{h_i - z - 1\, h_j - z - 1} \right). \tag{106}$$

Let us consider the case where both in $A$ and $B$ there is only a single spin ($\ell_A = \ell_B = 1$). In this case, using the definition given in Eq. (18) and remembering that non-zero contributions $|\mathcal{P}^{(\ell)}_{h_i h_j (z_i)}\rangle$ come from

$$\max\left(0, \lceil (h_i + h_j - \ell)/2 \rceil \right) \le z_i \le \min(h_i, h_j) \tag{107}$$

and that the difference of the heights is limited by $\ell$, namely $|h_i - h_j| \le \ell$, we have only the following possibilities for the states defined in $A$ and $B$

$$|\mathcal{P}^{(\ell_A = 1)}_{h_1 h_1 + 1 (z_1)}\rangle_{\bar{c}_1, \ldots, \bar{c}_{(h_1+1)}}^{c_{h_1}, \ldots, c_1} = |\uparrow^{\bar{c}_{(h_1+1)}}\rangle\, \delta_{z_1 h_1}\, \delta_{c_1, \bar{c}_1} \ldots \delta_{c_{h_1}, \bar{c}_{h_1}}, \tag{108}$$

$$|\mathcal{P}^{(\ell_A = 1)}_{h_1 h_1 - 1 (z_1)}\rangle_{\bar{c}_1, \ldots, \bar{c}_{(h_1-1)}}^{c_{h_1}, \ldots, c_1} = |\downarrow^{c_{h_1}}\rangle\, \delta_{z_1, h_1 - 1}\, \delta_{c_1, \bar{c}_1} \ldots \delta_{c_{(h_1-1)}, \bar{c}_{(h_1-1)}}, \tag{109}$$

$$|\mathcal{P}^{(\ell_B = 1)}_{h_4 - 1\, h_4 (z_3)}\rangle_{\hat{c}_1, \ldots, \hat{c}_{h_4}}^{\tilde{c}_{(h_4-1)}, \ldots, \tilde{c}_1} = |\uparrow^{\hat{c}_{h_4}}\rangle\, \delta_{z_3, h_4 - 1}\, \delta_{\tilde{c}_1, \hat{c}_1} \ldots \delta_{\tilde{c}_{(h_4-1)}, \hat{c}_{(h_4-1)}}, \tag{110}$$

$$|\mathcal{P}^{(\ell_B = 1)}_{h_4 + 1\, h_4 (z_3)}\rangle_{\hat{c}_1, \ldots, \hat{c}_{h_4}}^{\tilde{c}_{(h_4+1)}, \ldots, \tilde{c}_1} = |\downarrow^{\tilde{c}_{(h_4+1)}}\rangle\, \delta_{z_3 h_4}\, \delta_{\tilde{c}_1, \hat{c}_1} \ldots \delta_{\tilde{c}_{h_4}, \hat{c}_{h_4}}. \tag{111}$$

Eq. (109) is null for $h_1 = 0$ and Eq. (111) if $h_4 = 0$, however the coefficients in Eq. (104) take into account these possibilities when some heights in the argument of the Dyck numbers become negative. Calling $h = h_1$, $h' = h_4$, $z = z_2$ and the residual spin indices $\bar{c} = \bar{c}_{(h+1)}$ and $\tilde{c} = \tilde{c}_{(h'+1)}$ to simplify the notation, the ground state can be written as

$$\begin{aligned}
|\mathcal{P}^{(L)}\rangle &= \sum_{hh'z} \sum_{\substack{c_1 \ldots c_h \\ \hat{c}_1 \ldots \hat{c}_{h'}}} |\mathcal{P}^{(\ell_D)}_{0h}\rangle_{c_1, \ldots, c_h} \Big[ \sum_{\bar{c}} \sqrt{\mathcal{V}^{\uparrow\uparrow}_{hh'z}}\, |\uparrow^{\bar{c}}\rangle\, |\mathcal{P}^{(\ell_C)}_{h+1\, h'-1 (z)}\rangle_{\hat{c}_1, \ldots, \hat{c}_{(h'-1)}}^{\bar{c}, c_h, \ldots, c_1}\, |\uparrow^{\hat{c}_{h'}}\rangle \\
&\quad + \sum_{\bar{c}, \tilde{c}} \sqrt{\mathcal{V}^{\uparrow\downarrow}_{hh'z}}\, |\uparrow^{\bar{c}}\rangle\, |\mathcal{P}^{(\ell_C)}_{h+1\, h'+1 (z)}\rangle_{\hat{c}_1, \ldots, \hat{c}_{h'}, \tilde{c}}^{\bar{c}, c_h, \ldots, c_1}\, |\downarrow^{\tilde{c}}\rangle + \sqrt{\mathcal{V}^{\downarrow\uparrow}_{hh'z}}\, |\downarrow^{c_h}\rangle\, |\mathcal{P}^{(\ell_C)}_{h-1\, h'-1 (z)}\rangle_{\hat{c}_1, \ldots, \hat{c}_{(h'-1)}}^{c_{(h-1)}, \ldots, c_1}\, |\uparrow^{\hat{c}_{h'}}\rangle \\
&\quad + \sum_{\tilde{c}} \sqrt{\mathcal{V}^{\downarrow\downarrow}_{hh'z}}\, |\downarrow^{c_h}\rangle\, |\mathcal{P}^{(\ell_C)}_{h-1\, h'+1 (z)}\rangle_{\hat{c}_1, \ldots, \hat{c}_{h'}, \tilde{c}}^{c_{(h-1)}, \ldots, c_1}\, |\downarrow^{\tilde{c}}\rangle \Big] |\mathcal{P}^{(\ell_E)}_{h'0}\rangle_{\hat{c}_{h'}, \ldots, \hat{c}_1}, \tag{112}
\end{aligned}$$

where, from Eq. (105) and $\mathcal{L}^{(1)}_{h h+1} = \mathcal{D}^{(1)}_{01} = q$ together with $\mathcal{L}^{(1)}_{h h-1} = \mathcal{D}^{(1)}_{10} = 1$, we have

$$\mathcal{V}^{\uparrow\uparrow}_{hh'z} = \mathcal{A}_{\substack{h(h+1)(h'-1)h' \\ hz(h'-1)}} = \frac{\mathcal{D}^{(\ell_D)}_{0h} \left( \mathcal{D}^{(\ell_C)}_{h-z+1\, h'-z-1} - \mathcal{D}^{(\ell_C)}_{h-z\, h'-z-2} \right) \mathcal{D}^{(\ell_E)}_{h'0}}{\mathcal{D}^{(\ell_D + \ell_C + \ell_E + 2)}}\, q^{z - h - h' + 1}, \tag{113}$$

$$\mathcal{V}^{\uparrow\downarrow}_{hh'z} = \mathcal{A}_{\substack{h(h+1)(h'+1)h' \\ hzh'}} = \frac{\mathcal{D}^{(\ell_D)}_{0h} \left( \mathcal{D}^{(\ell_C)}_{h-z+1\, h'-z+1} - \mathcal{D}^{(\ell_C)}_{h-z\, h'-z} \right) \mathcal{D}^{(\ell_E)}_{h'0}}{\mathcal{D}^{(\ell_D + \ell_C + \ell_E + 2)}}\, q^{z - h - h' - 1}, \tag{114}$$

$$\mathcal{V}^{\downarrow\uparrow}_{hh'z} = \mathcal{A}_{\substack{h(h-1)(h'-1)h' \\ (h-1)z(h'-1)}} = \frac{\mathcal{D}^{(\ell_D)}_{0h} \left( \mathcal{D}^{(\ell_C)}_{h-z-1\, h'-z-1} - \mathcal{D}^{(\ell_C)}_{h-z-2\, h'-z-2} \right) \mathcal{D}^{(\ell_E)}_{h'0}}{\mathcal{D}^{(\ell_D + \ell_C + \ell_E + 2)}}\, q^{z - h - h' + 1}, \tag{115}$$

$$\mathcal{V}^{\downarrow\downarrow}_{hh'z} = \mathcal{A}_{\substack{h(h-1)(h'+1)h' \\ (h-1)zh'}} = \frac{\mathcal{D}^{(\ell_D)}_{0h} \left( \mathcal{D}^{(\ell_C)}_{h-z-1\, h'-z+1} - \mathcal{D}^{(\ell_C)}_{h-z-2\, h'-z} \right) \mathcal{D}^{(\ell_E)}_{h'0}}{\mathcal{D}^{(\ell_D + \ell_C + \ell_E + 2)}}\, q^{z - h - h' - 1}, \tag{116}$$

and where the sums over $h$ is limited by $\min(\ell_D, L - \ell_D)$, the sums over $h'$ is limited by $\min(\ell_E, L - \ell_E)$ and the sum over $z$ is defined differently for the four terms according to the

different heights of the borders of the central region as given by Eq. (107), where $z_i = z$, $\ell = \ell_C$, $h_i = h \pm 1$ and $h_j = h' \pm 1$. More importantly, we notice that in Eq. (112), for any $z$, the central states in the first and in the last term are orthogonal to any other while the central states in the second and third term can overlap, more explicitly any $(z + 2)$-th state of the second term coincides with the $z$-th state of the third term. These overlaps produce the coherent off-diagonal terms in the reduced density matrix $\rho_{AB}$.

In order to calculate the mutual information between $A$ and $B$ we have to calculate also $\rho_A$ and $\rho_B$. This can be done exploiting the tripartition of the ground state, Eq. (17) and Eq. (20),

$$\rho_A = \sum_{h\,c} \left( q^h \mathcal{A}^{(A)}_{h(h+1)h} |\uparrow^c\rangle \langle\uparrow^c| + q^{h-1} \mathcal{A}^{(A)}_{h(h-1)(h-1)} |\downarrow^c\rangle \langle\downarrow^c| \right), \tag{117}$$

$$\rho_B = \sum_{h\,c} \left( q^{h-1} \mathcal{A}^{(B)}_{(h-1)h(h-1)} |\uparrow^c\rangle \langle\uparrow^c| + q^h \mathcal{A}^{(B)}_{(h+1)hh} |\downarrow^c\rangle \langle\downarrow^c| \right), \tag{118}$$

where, using the same notation of Eq. (20),

$$\mathcal{A}^{(A)}_{h(h+1)h} = \frac{\mathcal{D}^{(\ell_D)}_{0h} \mathcal{D}^{(\ell_C+\ell_E+1)}_{h+1,0}}{\mathcal{D}^{(\ell_D+\ell_C+\ell_E+2)}} q^{-h}, \tag{119}$$

$$\mathcal{A}^{(A)}_{h(h-1)(h-1)} = \frac{\mathcal{D}^{(\ell_D)}_{0h} \mathcal{D}^{(\ell_C+\ell_E+1)}_{h-1,0}}{\mathcal{D}^{(\ell_D+\ell_C+\ell_E+2)}} q^{-h}, \tag{120}$$

$$\mathcal{A}^{(B)}_{(h-1)h(h-1)} = \frac{\mathcal{D}^{(\ell_D+\ell_C+1)}_{0h-1} \mathcal{D}^{(\ell_E)}_{h0}}{\mathcal{D}^{(\ell_D+\ell_C+\ell_E+2)}} q^{1-h}, \tag{121}$$

$$\mathcal{A}^{(B)}_{(h+1)hh} = \frac{\mathcal{D}^{(\ell_D+\ell_C+1)}_{0h+1} \mathcal{D}^{(\ell_E)}_{h0}}{\mathcal{D}^{(\ell_D+\ell_C+\ell_E+2)}} q^{-h-1}. \tag{122}$$

**Colorless case.** Let us consider for simplicity the colorless case ($q = 1$).

From Eq. (112) we can derive the reduced density matrix for the joint system $A \cup B$ after tracing out the rest of the chain

$$\rho_{AB} = \mathcal{V}^{\uparrow\uparrow} |\uparrow\rangle|\uparrow\rangle \langle\uparrow|\langle\uparrow| + \mathcal{V}^{\uparrow\downarrow} |\uparrow\rangle|\downarrow\rangle \langle\uparrow|\langle\downarrow| + \mathcal{V}^{\downarrow\uparrow} |\downarrow\rangle|\uparrow\rangle \langle\downarrow|\langle\uparrow| + \mathcal{V}^{\downarrow\downarrow} |\downarrow\rangle|\downarrow\rangle \langle\downarrow|\langle\downarrow| \tag{123}$$
$$+ \mathcal{V}^{X} \left( |\downarrow\rangle|\uparrow\rangle \langle\uparrow|\langle\downarrow| + |\uparrow\rangle|\downarrow\rangle \langle\downarrow|\langle\uparrow| \right),$$

which, on the basis $(|\uparrow\rangle|\uparrow\rangle, |\uparrow\rangle|\downarrow\rangle, |\downarrow\rangle|\uparrow\rangle, |\downarrow\rangle|\downarrow\rangle)^t$, can be written as

$$\rho_{AB} = \begin{pmatrix} \mathcal{V}^{\uparrow\uparrow} & 0 & 0 & 0 \\ 0 & \mathcal{V}^{\uparrow\downarrow} & \mathcal{V}^{X} & 0 \\ 0 & \mathcal{V}^{X} & \mathcal{V}^{\downarrow\uparrow} & 0 \\ 0 & 0 & 0 & \mathcal{V}^{\downarrow\downarrow} \end{pmatrix}, \tag{124}$$

where the coefficients, from Eqs. (113)-(116), are

$$\mathcal{V}^{\uparrow\uparrow} = \sum_{hh'z} \mathcal{V}^{\uparrow\uparrow}_{hh'z}, \quad \mathcal{V}^{\uparrow\downarrow} = \sum_{hh'z} \mathcal{V}^{\uparrow\downarrow}_{hh'z}, \quad \mathcal{V}^{\downarrow\uparrow} = \sum_{hh'z} \mathcal{V}^{\downarrow\uparrow}_{hh'z}, \quad \mathcal{V}^{\downarrow\downarrow} = \sum_{hh'z} \mathcal{V}^{\downarrow\downarrow}_{hh'z} \tag{125}$$

and the crossing term

$$\mathcal{V}^{X} = \sum_{hh'z} \sqrt{\mathcal{V}^{\uparrow\downarrow}_{hh'z+2} \mathcal{V}^{\downarrow\uparrow}_{hh'z}} = \sum_{hh'z} \mathcal{V}^{\downarrow\uparrow}_{hh'z} = \mathcal{V}^{\downarrow\uparrow}, \tag{126}$$

which is actually at the origin of the coherence and of the long-distance entanglement between the spins. One can verify through Eqs. (113)-(116) that

$$\mathrm{Tr}(\rho_{AB}) = \mathcal{V}^{\uparrow\uparrow} + \mathcal{V}^{\uparrow\downarrow} + \mathcal{V}^{\downarrow\uparrow} + \mathcal{V}^{\downarrow\downarrow} = 1. \tag{127}$$

Eq. (123) is indeed the generalization of Eq. (57) and reduces to it for $\ell_D = \ell_E = 1$ since the first and he last spins are fixed to be up and down respectively. The eigenvalues of $\rho_{AB}$ are $\mathcal{V}^{\uparrow\uparrow}$, $\mathcal{V}^{\downarrow\downarrow}$, and

$$\mathcal{V}^{\pm} \equiv \frac{1}{2}\left(\mathcal{V}^{\uparrow\downarrow} + \mathcal{V}^{\downarrow\uparrow} \pm \sqrt{4(\mathcal{V}^X)^2 + (\mathcal{V}^{\uparrow\downarrow} - \mathcal{V}^{\downarrow\uparrow})^2}\right). \tag{128}$$

From Eq. (123) and the following reduced density matrices for the regions $A$ and $B$

$$\rho_A = \mathcal{A}^{\uparrow} |\uparrow\rangle\langle\uparrow| + \mathcal{A}^{\downarrow} |\downarrow\rangle\langle\downarrow| \tag{129}$$

$$\rho_B = \mathcal{B}^{\uparrow} |\uparrow\rangle\langle\uparrow| + \mathcal{B}^{\downarrow} |\downarrow\rangle\langle\downarrow|, \tag{130}$$

where the coefficients, using Eqs. (119)-(122), are

$$\mathcal{A}^{\uparrow} = \sum_h \mathcal{A}^{(A)}_{h(h+1)h}, \quad \mathcal{A}^{\downarrow} = \sum_h \mathcal{A}^{(A)}_{h(h-1)(h-1)}, \quad \mathcal{B}^{\uparrow} = \sum_h \mathcal{A}^{(B)}_{(h-1)h(h-1)}, \quad \mathcal{B}^{\downarrow} = \sum_h \mathcal{A}^{(B)}_{(h+1)hh}. \tag{131}$$

We can calculate the corresponding entanglement entropies and then the mutual information, $\mathcal{I}_{AB} = S_A + S_B - S_{AB}$ exactly, by means of Eqs. (113)-(116), (119)-(122), (125), (126), (128), (164), which is

$$\mathcal{I}_{AB} = \mathcal{V}^{\uparrow\uparrow} \log \mathcal{V}^{\uparrow\uparrow} + \mathcal{V}^{\downarrow\downarrow} \log \mathcal{V}^{\downarrow\downarrow} + \mathcal{V}^{+} \log \mathcal{V}^{+} + \mathcal{V}^{-} \log \mathcal{V}^{-}$$
$$- \mathcal{A}^{\uparrow} \log \mathcal{A}^{\uparrow} - \mathcal{A}^{\downarrow} \log \mathcal{A}^{\downarrow} - \mathcal{B}^{\uparrow} \log \mathcal{B}^{\uparrow} - \mathcal{B}^{\downarrow} \log \mathcal{B}^{\downarrow}. \tag{132}$$

Explicit calculation shows that also for very large system size $L$ the mutual information between two spins in the bulk does not vanish, as shown in the left panel of Fig. 11. Actually, its asymptotic value increases when considering two spins located more and more deeply in the bulk (see right panel of Fig. 11). Actually we expected and verified that increasing $\ell_D$ and

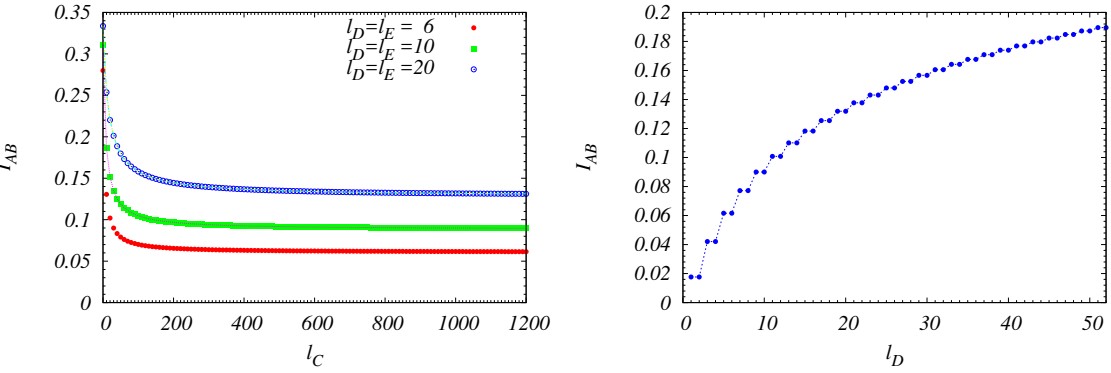

Figure 11: (Left) Mutual information between two spins in the bulk of a colorless Fredkin chain at distances $\ell_D = \ell_E = 6, 10, 20$ from the edges, as a function of the relative distance $\ell_C$; (Right) Mutual information between two spins in the bulk as a function of the distance $\ell_D = \ell_E$ from the edges for colorless Fredkin chain with size $L = 1000$.

$\ell_E$, so going in the deep bulk, the probabilities of getting $\uparrow$ and $\uparrow$ ($\mathcal{V}^{\uparrow\uparrow}$) or $\uparrow$ and $\downarrow$ ($\mathcal{V}^{\uparrow\downarrow}$) or $\downarrow$ and $\uparrow$ ($\mathcal{V}^{\downarrow\uparrow}$) or finally $\downarrow$ and $\downarrow$ ($\mathcal{V}^{\downarrow\downarrow}$) should become the same and equal to 1/4, since also the probabilities of having one spin $\uparrow$ or $\downarrow$, both in $A$ and $B$, should be 1/2. This means that

$$\rho_{AB} \to \frac{1}{4}\begin{pmatrix} 1 & 0 & 0 & 0 \\ 0 & 1 & 1 & 0 \\ 0 & 1 & 1 & 0 \\ 0 & 0 & 0 & 1 \end{pmatrix}, \quad \rho_A \to \frac{1}{2}\begin{pmatrix} 1 & 0 \\ 0 & 1 \end{pmatrix}, \quad \rho_B \to \frac{1}{2}\begin{pmatrix} 1 & 0 \\ 0 & 1 \end{pmatrix}, \tag{133}$$

and consequently, for large system size $L$ and for $\ell_D, \ell_E \gg 1$, deeply in the bulk,

$$\mathcal{I}_{AB} \to \frac{1}{2}\log 2, \tag{134}$$

which is also the same value, $\mathcal{I}_{AB} \approx 0.35$, obtained for $\ell_C = 0$ and $\ell_D, \ell_E \gg 1$, as shown by the first points in the left panel of Fig. 11). Eq. (134) is actually the asymptotic value of the curve in the right panel of Fig. 11. The staircase effect observed in Fig. 11 (right panel) is due to the fact that the numbers $\mathcal{D}_{hh'}^{(\ell)}$ are zero for $(\ell + h + h')$ odd integers. This feature is also present in other ground-state quantities like the magnetization and the correlation functions [9, 23].

## 6.2 Motzkin model

For the integer spin model the coefficients appearing in Eq. (104) are the following

$$\mathcal{A}_{\substack{h_1 h_2 h_3 h_4 \\ z_1 z_2 z_3}} = \frac{\mathcal{M}_{0h_1}^{(\ell_D)} \mathcal{L}_{h_1 h_2 z_1}^{(\ell_A)} \mathcal{L}_{h_2 h_3 z_2}^{(\ell_C)} \mathcal{L}_{h_3 h_4 z_3}^{(\ell_B)} \mathcal{M}_{h_4 0}^{(\ell_E)}}{\mathcal{M}^{(L)}} q^{z_1 + z_2 + z_3 - (h_1 + h_2 + h_3 + h_4)}, \tag{135}$$

where

$$\mathcal{L}_{h_i h_j z}^{(\ell)} = \left( \mathcal{M}_{h_i - z\, h_j - z}^{(\ell)} - \mathcal{M}_{h_i - z - 1\, h_j - z - 1}^{(\ell)} \right). \tag{136}$$

As done for the Fredkin chain, let us consider the case where both in $A$ and $B$ there is only a single spin ($\ell_A = \ell_B = 1$). Proceeding analogously as done for the half-integer case, using Eq. (104) we get, for the ground state of the Motzkin model, the following decomposition

$$
\begin{aligned}
|\mathcal{P}^{(L)}\rangle &= \sum_{hh'z} \sum_{\substack{c_1 \dots c_h \\ \hat{c}_1 \dots \hat{c}_{h'}}} |\mathcal{P}_{0h}^{(\ell_D)}\rangle_{c_1,\dots,c_h} \Big[ \sum_{\bar{c}} \sqrt{\mathcal{V}_{hh'z}^{\Uparrow\Uparrow}} \, |\Uparrow^{\bar{c}}\rangle \, |\mathcal{P}_{h+1\,h'-1\,(z)}^{(\ell_C)}\rangle_{\substack{\bar{c},c_h,\dots,c_1 \\ \hat{c}_1,\dots,\hat{c}_{(h'-1)}}} |\Uparrow^{\hat{c}_{h'}}\rangle \\
&\quad + \sum_{\bar{c},\tilde{c}} \sqrt{\mathcal{V}_{hh'z}^{\Uparrow\Downarrow}} \, |\Uparrow^{\bar{c}}\rangle \, |\mathcal{P}_{h+1\,h'+1\,(z)}^{(\ell_C)}\rangle_{\substack{\bar{c},c_h,\dots,c_1 \\ \hat{c}_1,\dots,\hat{c}_{h'},\tilde{c}}} |\Downarrow^{\tilde{c}}\rangle + \sqrt{\mathcal{V}_{hh'z}^{\Downarrow\Uparrow}} \, |\Downarrow^{c_h}\rangle \, |\mathcal{P}_{h-1\,h'-1\,(z)}^{(\ell_C)}\rangle_{\substack{c_{(h-1)},\dots,c_1 \\ \hat{c}_1,\dots,\hat{c}_{(h'-1)}}} |\Uparrow^{\hat{c}_{h'}}\rangle \\
&\quad + \sum_{\tilde{c}} \sqrt{\mathcal{V}_{hh'z}^{\Downarrow\Downarrow}} \, |\Downarrow^{c_h}\rangle \, |\mathcal{P}_{h-1\,h'+1\,(z)}^{(\ell_C)}\rangle_{\substack{c_{(h-1)},\dots,c_1 \\ \hat{c}_1,\dots,\hat{c}_{h'},\tilde{c}}} |\Downarrow^{\tilde{c}}\rangle + \sum_{\bar{c}} \sqrt{\mathcal{V}_{hh'z}^{\Uparrow 0}} \, |\Uparrow^{\bar{c}}\rangle \, |\mathcal{P}_{h+1\,h'\,(z)}^{(\ell_C)}\rangle_{\substack{\bar{c},c_h,\dots,c_1 \\ \hat{c}_1,\dots,\hat{c}_{h'}}} |0\rangle \\
&\quad + \sqrt{\mathcal{V}_{hh'z}^{\Downarrow 0}} \, |\Downarrow^{c_h}\rangle \, |\mathcal{P}_{h-1\,h'\,(z)}^{(\ell_C)}\rangle_{\substack{c_{(h-1)},\dots,c_1 \\ \hat{c}_1,\dots,\hat{c}_{h'}}} |0\rangle + \sqrt{\mathcal{V}_{hh'z}^{0\Uparrow}} \, |0\rangle \, |\mathcal{P}_{h\,h'-1\,(z)}^{(\ell_C)}\rangle_{\substack{c_h,\dots,c_1 \\ \hat{c}_1,\dots,\hat{c}_{(h'-1)}}} |\Uparrow^{\hat{c}_{h'}}\rangle \\
&\quad + \sum_{\tilde{c}} \sqrt{\mathcal{V}_{hh'z}^{0\Downarrow}} \, |0\rangle \, |\mathcal{P}_{h\,h'+1\,(z)}^{(\ell_C)}\rangle_{\substack{c_h,\dots,c_1 \\ \hat{c}_1,\dots,\hat{c}_{h'},\tilde{c}}} |\Downarrow^{\tilde{c}}\rangle + \sqrt{\mathcal{V}_{hh'z}^{00}} \, |0\rangle \, |\mathcal{P}_{h\,h'\,(z)}^{(\ell_C)}\rangle_{\substack{c_h,\dots,c_1 \\ \hat{c}_1,\dots,\hat{c}_{h'}}} |0\rangle \Big] |\mathcal{P}_{h'0}^{(\ell_E)}\rangle_{\hat{c}_{h'},\dots,\hat{c}_1}, \tag{137}
\end{aligned}
$$

where the coefficients, according to Eq. (135), since $\mathcal{L}_{hh+1}^{(1)} = \mathcal{M}_{01}^{(1)} = q$ and $\mathcal{L}_{hh}^{(1)} = \mathcal{M}_{00}^{(1)} = \mathcal{L}_{hh-1}^{(1)} = \mathcal{M}_{10}^{(1)} = 1$, are

$$\mathcal{V}_{hh'z}^{\Uparrow\Uparrow\Uparrow} = \mathcal{A}_{\substack{h(h+1)(h'-1)h' \\ hz(h'-1)}} = \frac{\mathcal{M}_{0h}^{(\ell_D)}\left(\mathcal{M}_{h-z+1\,h'-z-1}^{(\ell_C)} - \mathcal{M}_{h-z\,h'-z-2}^{(\ell_C)}\right)\mathcal{M}_{h'0}^{(\ell_E)}}{\mathcal{M}^{(\ell_D+\ell_C+\ell_E+2)}}q^{z-h-h'+1} \tag{138}$$

$$\mathcal{V}_{hh'z}^{\Uparrow\Uparrow\Downarrow} = \mathcal{A}_{\substack{h(h+1)(h'+1)h' \\ hzh'}} = \frac{\mathcal{M}_{0h}^{(\ell_D)}\left(\mathcal{M}_{h-z+1\,h'-z+1}^{(\ell_C)} - \mathcal{M}_{h-z\,h'-z}^{(\ell_C)}\right)\mathcal{M}_{h'0}^{(\ell_E)}}{\mathcal{M}^{(\ell_D+\ell_C+\ell_E+2)}}q^{z-h-h'-1} \tag{139}$$

$$\mathcal{V}_{hh'z}^{\Downarrow\Uparrow} = \mathcal{A}_{\substack{h(h-1)(h'-1)h' \\ (h-1)z(h'-1)}} = \frac{\mathcal{M}_{0h}^{(\ell_D)}\left(\mathcal{M}_{h-z-1\,h'-z-1}^{(\ell_C)} - \mathcal{M}_{h-z-2\,h'-z-2}^{(\ell_C)}\right)\mathcal{M}_{h'0}^{(\ell_E)}}{\mathcal{M}^{(\ell_D+\ell_C+\ell_E+2)}}q^{z-h-h'+1} \tag{140}$$

$$\mathcal{V}_{hh'z}^{\Downarrow\Downarrow} = \mathcal{A}_{\substack{h(h-1)(h'+1)h' \\ (h-1)zh'}} = \frac{\mathcal{M}_{0h}^{(\ell_D)}\left(\mathcal{M}_{h-z-1\,h'-z+1}^{(\ell_C)} - \mathcal{M}_{h-z-2\,h'-z}^{(\ell_C)}\right)\mathcal{M}_{h'0}^{(\ell_E)}}{\mathcal{M}^{(\ell_D+\ell_C+\ell_E+2)}}q^{z-h-h'-1} \tag{141}$$

$$\mathcal{V}_{hh'z}^{\Uparrow 0} = \mathcal{A}_{\substack{h(h+1)h'h' \\ hzh'}} = \frac{\mathcal{M}_{0h}^{(\ell_D)}\left(\mathcal{M}_{h-z+1\,h'-z}^{(\ell_C)} - \mathcal{M}_{h-z\,h'-z-1}^{(\ell_C)}\right)\mathcal{M}_{h'0}^{(\ell_E)}}{\mathcal{M}^{(\ell_D+\ell_C+\ell_E+2)}}q^{z-h-h'} \tag{142}$$

$$\mathcal{V}_{hh'z}^{\Downarrow 0} = \mathcal{A}_{\substack{h(h-1)h'h' \\ (h-1)zh'}} = \frac{\mathcal{M}_{0h}^{(\ell_D)}\left(\mathcal{M}_{h-z-1\,h'-z}^{(\ell_C)} - \mathcal{M}_{h-z-2\,h'-z-1}^{(\ell_C)}\right)\mathcal{M}_{h'0}^{(\ell_E)}}{\mathcal{M}^{(\ell_D+\ell_C+\ell_E+2)}}q^{z-h-h'} \tag{143}$$

$$\mathcal{V}_{hh'z}^{0\Uparrow} = \mathcal{A}_{\substack{hh(h'-1)h' \\ hz(h'-1)}} = \frac{\mathcal{M}_{0h}^{(\ell_D)}\left(\mathcal{M}_{h-z\,h'-z-1}^{(\ell_C)} - \mathcal{M}_{h-z-1\,h'-z-2}^{(\ell_C)}\right)\mathcal{M}_{h'0}^{(\ell_E)}}{\mathcal{M}^{(\ell_D+\ell_C+\ell_E+2)}}q^{z-h-h'+1} \tag{144}$$

$$\mathcal{V}_{hh'z}^{0\Downarrow} = \mathcal{A}_{\substack{hh(h'+1)h' \\ hzh'}} = \frac{\mathcal{M}_{0h}^{(\ell_D)}\left(\mathcal{M}_{h-z\,h'-z+1}^{(\ell_C)} - \mathcal{M}_{h-z-1\,h'-z}^{(\ell_C)}\right)\mathcal{M}_{h'0}^{(\ell_E)}}{\mathcal{M}^{(\ell_D+\ell_C+\ell_E+2)}}q^{z-h-h'-1} \tag{145}$$

$$\mathcal{V}_{hh'z}^{00} = \mathcal{A}_{\substack{hhh'h' \\ hzh'}} = \frac{\mathcal{M}_{0h}^{(\ell_D)}\left(\mathcal{M}_{h-z\,h'-z}^{(\ell_C)} - \mathcal{M}_{h-z-1\,h'-z-1}^{(\ell_C)}\right)\mathcal{M}_{h'0}^{(\ell_E)}}{\mathcal{M}^{(\ell_D+\ell_C+\ell_E+2)}}q^{z-h-h'}. \tag{146}$$

On the other hand the reduced density matrix of a single spin in $A$ and $B$ can be determined performing the tripartition described by Eq. (17)

$$\rho_A = \sum_h \mathcal{A}_{hhh}^{(A)}|0\rangle\langle 0| + \sum_{hc}\left(q^h \mathcal{A}_{h(h+1)h}^{(A)}|\Uparrow^c\rangle\langle\Uparrow^c| + q^{h-1}\mathcal{A}_{h(h-1)(h-1)}^{(A)}|\Downarrow^c\rangle\langle\Downarrow^c|\right), \tag{147}$$

$$\rho_B = \sum_h \mathcal{A}_{hhh}^{(B)}|0\rangle\langle 0| + \sum_{hc}\left(q^{h-1}\mathcal{A}_{(h-1)h(h-1)}^{(B)}|\Uparrow^c\rangle\langle\Uparrow^c| + q^h \mathcal{A}_{(h+1)hh}^{(B)}|\Downarrow^c\rangle\langle\Downarrow^c|\right), \tag{148}$$

where, using the definitions reported in Eq. (67), the coefficients are

$$\mathcal{A}_{hhh}^{(A)} = \frac{\mathcal{M}_{0h}^{(\ell_D)}\mathcal{M}_{h0}^{(\ell_C+\ell_E+1)}}{\mathcal{M}^{(\ell_D+\ell_C+\ell_E+2)}}q^{-h}, \tag{149}$$

$$\mathcal{A}_{h(h+1)h}^{(A)} = \frac{\mathcal{M}_{0h}^{(\ell_D)}\mathcal{M}_{h+1,0}^{(\ell_C+\ell_E+1)}}{\mathcal{M}^{(\ell_D+\ell_C+\ell_E+2)}}q^{-h}, \tag{150}$$

$$\mathcal{A}_{h(h-1)(h-1)}^{(A)} = \frac{\mathcal{M}_{0h}^{(\ell_D)}\mathcal{M}_{h-1,0}^{(\ell_C+\ell_E+1)}}{\mathcal{M}^{(\ell_D+\ell_C+\ell_E+2)}}q^{-h}, \tag{151}$$

$$\mathcal{A}_{hhh}^{(B)} = \frac{\mathcal{M}_{0h}^{(\ell_D+\ell_C+1)}\mathcal{M}_{h0}^{(\ell_E)}}{\mathcal{M}^{(\ell_D+\ell_C+\ell_E+2)}}q^{-h}, \tag{152}$$

$$\mathcal{A}_{(h-1)h(h-1)}^{(B)} = \frac{\mathcal{M}_{0h-1}^{(\ell_D+\ell_C+1)}\mathcal{M}_{h0}^{(\ell_E)}}{\mathcal{M}^{(\ell_D+\ell_C+\ell_E+2)}}q^{1-h}, \tag{153}$$

$$\mathcal{A}_{(h+1)hh}^{(B)} = \frac{\mathcal{M}_{0h+1}^{(\ell_D+\ell_C+1)}\mathcal{M}_{h0}^{(\ell_E)}}{\mathcal{M}^{(\ell_D+\ell_C+\ell_E+2)}}q^{-h-1}. \tag{154}$$

**Colorless case.** As done for Fredkin chain, let us consider for simplicity the colorless Motzkin model ($q = 1$).

From Eq. (138) we can derive the reduced density matrix for the joint system $A \cup B$ after tracing out the rest of the chain

$$
\begin{aligned}
\rho_{AB} ={}& \mathcal{V}^{\Uparrow\Uparrow} |\Uparrow\rangle|\Uparrow\rangle \langle\Uparrow|\langle\Uparrow| + \mathcal{V}^{\Uparrow 0} |\Uparrow\rangle|0\rangle \langle\Uparrow|\langle 0| + \mathcal{V}^{0\Uparrow} |0\rangle|\Uparrow\rangle \langle 0|\langle\Uparrow| + \mathcal{V}^{X_u}\Big( |\Uparrow\rangle|0\rangle \langle 0|\langle\Uparrow| + |0\rangle|\Uparrow\rangle \langle\Uparrow|\langle 0| \Big) \\
&+\mathcal{V}^{\Uparrow\Downarrow} |\Uparrow\rangle|\Downarrow\rangle \langle\Uparrow|\langle\Downarrow| + \mathcal{V}^{00} |0\rangle|0\rangle \langle 0|\langle 0| + \mathcal{V}^{\Downarrow\Uparrow} |\Downarrow\rangle|\Uparrow\rangle \langle\Downarrow|\langle\Uparrow| + \mathcal{V}^{X_0^{(1)}}\Big( |\Uparrow\rangle|\Downarrow\rangle \langle\Downarrow|\langle\Uparrow| + |\Downarrow\rangle|\Uparrow\rangle \langle\Uparrow|\langle\Downarrow| \Big) \\
&+\mathcal{V}^{X_0^{(2)}}\Big( |\Uparrow\rangle|\Downarrow\rangle \langle 0|\langle 0| + |0\rangle|0\rangle \langle\Uparrow|\langle\Downarrow| \Big) + \mathcal{V}^{X_0^{(3)}}\Big( |0\rangle|0\rangle \langle\Downarrow|\langle\Uparrow| + |\Downarrow\rangle|\Uparrow\rangle \langle 0|\langle 0| \Big) + \mathcal{V}^{0\Downarrow} |0\rangle|\Downarrow\rangle \langle 0|\langle\Downarrow| \\
&+\mathcal{V}^{\Downarrow 0} |\Downarrow\rangle|0\rangle \langle\Downarrow|\langle 0| + \mathcal{V}^{X_d}\Big( |0\rangle|\Downarrow\rangle \langle\Downarrow|\langle 0| + |\Downarrow\rangle|0\rangle \langle 0|\langle\Downarrow| \Big) + \mathcal{V}^{\Downarrow\Downarrow} |\Downarrow\rangle|\Downarrow\rangle \langle\Downarrow|\langle\Downarrow| ,
\end{aligned}
\tag{155}
$$

where, denoting $\sigma, \sigma' = \Uparrow, 0, \Downarrow$, the coefficients are defined by

$$
\mathcal{V}^{\sigma\sigma'} = \sum_{hh'z} \mathcal{V}^{\sigma\sigma'}_{hh'z},
\tag{156}
$$

$$
\mathcal{V}^{X_u} = \sum_{hh'z} \sqrt{\mathcal{V}^{\Uparrow 0}_{hh'z+1} \mathcal{V}^{0\Uparrow}_{hh'z}} = \mathcal{V}^{0\Uparrow} ,
\tag{157}
$$

$$
\mathcal{V}^{X_0^{(1)}} = \sum_{hh'z} \sqrt{\mathcal{V}^{\Uparrow\Downarrow}_{hh'z+2} \mathcal{V}^{\Downarrow\Uparrow}_{hh'z}} = \mathcal{V}^{\Downarrow\Uparrow} ,
\tag{158}
$$

$$
\mathcal{V}^{X_0^{(2)}} = \sum_{hh'z} \sqrt{\mathcal{V}^{\Uparrow\Downarrow}_{hh'z+1} \mathcal{V}^{00}_{hh'z}} = \mathcal{V}^{00} ,
\tag{159}
$$

$$
\mathcal{V}^{X_0^{(3)}} = \sum_{hh'z} \sqrt{\mathcal{V}^{\Downarrow\Uparrow}_{hh'z} \mathcal{V}^{00}_{hh'z+1}} = \mathcal{V}^{\Downarrow\Uparrow} .
\tag{160}
$$

Choosing an opportune basis, the reduced density matrix can be written in a block diagonal form as it follows

$$
\rho_{AB} =
\begin{pmatrix}
\mathcal{V}^{\Uparrow\Uparrow} & 0 & 0 & 0 & 0 & 0 & 0 & 0 & 0 \\
0 & \mathcal{V}^{\Uparrow 0} & \mathcal{V}^{0\Uparrow} & 0 & 0 & 0 & 0 & 0 & 0 \\
0 & \mathcal{V}^{0\Uparrow} & \mathcal{V}^{0\Uparrow} & 0 & 0 & 0 & 0 & 0 & 0 \\
0 & 0 & 0 & \mathcal{V}^{\Uparrow\Downarrow} & \mathcal{V}^{00} & \mathcal{V}^{\Downarrow\Uparrow} & 0 & 0 & 0 \\
0 & 0 & 0 & \mathcal{V}^{00} & \mathcal{V}^{00} & \mathcal{V}^{\Downarrow\Uparrow} & 0 & 0 & 0 \\
0 & 0 & 0 & \mathcal{V}^{\Downarrow\Uparrow} & \mathcal{V}^{\Downarrow\Uparrow} & \mathcal{V}^{\Downarrow\Uparrow} & 0 & 0 & 0 \\
0 & 0 & 0 & 0 & 0 & 0 & \mathcal{V}^{\Downarrow 0} & \mathcal{V}^{\Downarrow 0} & 0 \\
0 & 0 & 0 & 0 & 0 & 0 & \mathcal{V}^{0\Downarrow} & \mathcal{V}^{0\Downarrow} & 0 \\
0 & 0 & 0 & 0 & 0 & 0 & 0 & 0 & \mathcal{V}^{\Downarrow\Downarrow}
\end{pmatrix}.
\tag{161}
$$

We verified that $\text{Tr}(\rho_{AB}) = 1$. Now, together with the reduced density matrices for the single spins in $A$ and $B$

$$
\rho_A = \mathcal{A}^{\Uparrow} |\Uparrow\rangle \langle\Uparrow| + \mathcal{A}^0 |0\rangle \langle 0| + \mathcal{A}^{\Downarrow} |\Downarrow\rangle \langle\Downarrow| ,
\tag{162}
$$

$$
\rho_B = \mathcal{B}^{\Uparrow} |\Uparrow\rangle \langle\Uparrow| + \mathcal{B}^0 |0\rangle \langle 0| + \mathcal{B}^{\Downarrow} |\Downarrow\rangle \langle\Downarrow| ,
\tag{163}
$$

where the coefficients, related to Eqs. (149)-(154), are

$$
\mathcal{A}^{\Uparrow} = \sum_h \mathcal{A}^{(A)}_{h(h+1)h}, \quad \mathcal{A}^0 = \sum_h \mathcal{A}^{(A)}_{hhh}, \quad \mathcal{A}^{\Downarrow} = \sum_h \mathcal{A}^{(A)}_{h(h-1)(h-1)},
\tag{164}
$$

$$
\mathcal{B}^{\Uparrow} = \sum_h \mathcal{A}^{(B)}_{(h-1)h(h-1)}, \quad \mathcal{B}^0 = \sum_h \mathcal{A}^{(B)}_{hhh}, \quad \mathcal{B}^{\Downarrow} = \sum_h \mathcal{A}^{(B)}_{(h+1)hh},
\tag{165}
$$

we can calculate the entanglement entropies and finally the mutual information, $\mathcal{I}_{AB} = S_A + S_B - S_{AB}$, exactly. Examples of the exact results for the mutual information shared by two spins in the bulk have been given in Fig. 12 where we show that $\mathcal{I}_{AB}$ does not vanish when the distance of the spins in the bulk goes to zero but it saturates at some values which increase going more and more deeply into the bulk. Increasing the distances from the edges,

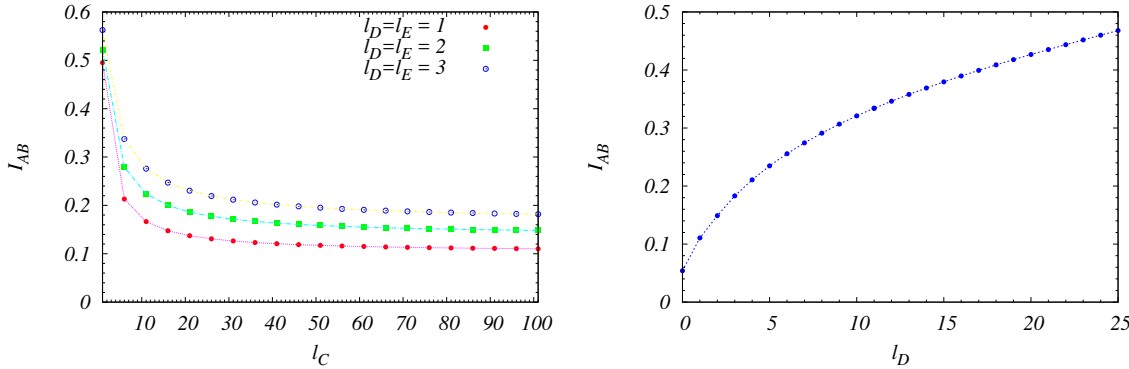

Figure 12: (Left) Mutual information between two spins in the bulk of a colorless Motzkin chain at distances $\ell_D = \ell_E = 1, 2, 3$ from the edges, as a function of the relative distance $\ell_C$; (Right) Mutual information between two spins in the bulk as a function of the distance $\ell_D = \ell_E$ from the edges for colorless Motzkin chain with size $L = 100$.

$\ell_D$ and $\ell_E$, we expected and verified that the probabilities in $\rho_{AB}$ factorize

$$\mathcal{V}^{\sigma\sigma'} \rightarrow \mathcal{A}^{\sigma}\mathcal{B}^{\sigma'}, \tag{166}$$

and, upon further increasing $\ell_D$ and $\ell_E$, namely very deeply in the bulk, they become homogeneous, $\mathcal{V}^{\sigma\sigma'} \rightarrow \frac{1}{9}$ and $\mathcal{A}^{\sigma} \rightarrow \frac{1}{3}$, $\mathcal{B}^{\sigma'} \rightarrow \frac{1}{3}$. The eigenvalues of $\rho_{AB}$, in this limit, become $1/3, 2/9, 2/9, 1/9, 1/9, 0, 0, 0, 0$, so that the entanglement entropy is $S_{AB} \rightarrow \frac{5}{3}\log 3 - \frac{4}{9}\log 2$. As a result the asymptotic upper bound for the mutual information between two spins in the bulk is given by

$$\mathcal{I}_{AB} \rightarrow \frac{1}{3}\log 3 + \frac{4}{9}\log 2. \tag{167}$$

## 6.3 General results for any disjoint intervals

The physical explanation of what seen so far, at least for the colorless cases, is the following. Any state defined on a segment of the chain, $|\mathcal{P}_{hh'(z)}\rangle$, can be defined by two quantum numbers:
*i)* the magnetization $\mathsf{m} = (h' - h)$ (the number of up-spins minus the number of down-spins),
*ii)* the horizon $z$ (the lowest level of the paths),
so that we can write $|\mathcal{P}_{hh'(z)}\rangle = |\mathsf{m}, z\rangle$. Considering the decomposition as depicted in Fig. 10, we have that the ground state written in Eq. (104) can be rewritten schematically as

$$|\mathcal{P}^{(L)}\rangle = \sum_{\{\mathsf{m}\}\{z\}} \sqrt{\mathcal{A}(\mathsf{m}, z)}\, |\mathsf{m}_D, 0\rangle\, |\mathsf{m}_A, z_A\rangle\, |\mathsf{m}_C, z_C\rangle\, |\mathsf{m}_B, z_B\rangle\, |\mathsf{m}_E, 0\rangle, \tag{168}$$

where the sum is such that $\mathsf{m}_D + \mathsf{m}_A + \mathsf{m}_C + \mathsf{m}_B + \mathsf{m}_E = 0$ and is restricted by the request that, for any set of magnetizations, one has to get a Fredkin or a Motzkin path (therefore, $\mathsf{m}_D$ has to be non-negative, as well as any initial sums, for instance $\mathsf{m}_D + \mathsf{m}_A + ..$, as a consequence $\mathsf{m}_E$

has to be non-positive). The reduced density matrix of $A \cup C \cup B$, after tracing over $D$ and $E$, is

$$
\begin{aligned}
\rho_{ACB} &= \mathrm{Tr}_{DE} |\mathcal{P}^{(L)}\rangle \langle \mathcal{P}^{(L)}| \\
&= \sum_{\substack{\{m,m'\} \\ \{z,z'\}}} \sqrt{\mathcal{A}(m,z)\mathcal{A}(m',z')} \, |m_A, z_A\rangle |m_C, z_C\rangle |m_B, z_B\rangle \langle m'_A, z'_A| \langle m'_C, z'_C| \langle m'_B, z'_B|,
\end{aligned}
\tag{169}
$$

where the central magnetizations are restricted by

$$
m_C = -(m_D + m_E + m_A + m_B), \tag{170}
$$
$$
m'_C = -(m_D + m_E + m'_A + m'_B), \tag{171}
$$

therefore

$$
\langle m'_C, z'_C | m_C, z_C \rangle = \delta_{(m'_A + m'_B),(m_A + m_B)} \, \delta_{z'_C, z_C} \tag{172}
$$

namely, the overlap is one if the total magnetizations in $A$ and $B$ are the same. This constraint implies that there are coherent terms in the reduced density matrix for $A$ and $B$ after integrating over $C$. As a result the reduced density matrix $\rho_{AB}$ can be written in a block diagonal form, where each block is defined by a magnetization sector,

$$
\rho_{AB} = \begin{pmatrix} \boxed{\mathbb{V}_{\ell_S}} & & & & \\ & \ddots & & & \\ & & \boxed{\mathbb{V}_{\ell_S - i}} & & \\ & & & \ddots & \\ & & & & \boxed{\mathbb{V}_{-\ell_S}} \end{pmatrix}. \tag{173}
$$

The blocks $\mathbb{V}_{\ell_S - i}$ are square matrices defined by the maximum total magnetization

$$
\ell_S = \max(m_A + m_B) = \ell_A + \ell_B \tag{174}
$$

and the index $i$ which runs differently for the integer or half-integer cases, i.e. $i = 0, 2, 4, \ldots, 2\ell_S$, for the Fredkin model, and $i = 0, 1, 2, \ldots, 2\ell_S$, for the Motzkin model, namely there are $(\ell_S + 1)$ square blocks for the Fredkin case and $(2\ell_S + 1)$ blocks for the Motzkin one. The dimension of $\rho_{AB}$ is $[(\ell_A + 1)(\ell_B + 1)] \times [(\ell_A + 1)(\ell_B + 1)]$ for the Fredkin model and $[(2\ell_A + 1)(2\ell_B + 1)] \times [(2\ell_A + 1)(2\ell_B + 1)]$ for the Motzkin model.

The dimension of the blocks $\mathbb{V}_{\ell_S}$ and $\mathbb{V}_{-\ell_S}$ is 1, while the dimensions of the other blocks are larger for sectors with smaller modulus of the spin. In general terms, $\mathbb{V}_n$ is a $d_n \times d_n$ matrix, with

$$
d_n = \sum_{h=-\ell_A}^{\ell_A} \Theta[\ell_B + n - h]\Theta[\ell_B - n + h], \tag{175}
$$

where the sum runs over $h$ with step 1 for the Motzkin and step 2 for the Fredkin, and $\Theta[n] = 1$ for $n \geq 0$ and $\Theta[n] = 0$ otherwise. Defining

$$
\mathcal{V}^{m_A m_B}_{hh' z_A z_B z} = \frac{\mathcal{D}^{(\ell_D)}_{0h} \mathcal{L}^{(\ell_A)}_{h(h+m_A)z_A} \mathcal{L}^{(\ell_C)}_{(h+m_A)(h'-m_B)z} \mathcal{L}^{(\ell_B)}_{(h'-m_B)h' z_B} \mathcal{D}^{(\ell_E)}_{h'0}}{\mathcal{D}^{(\ell_D + \ell_A + \ell_C + \ell_B + \ell_E)}}, \tag{176}
$$

where $\mathcal{L}^{\ell}_{hh'z}$ is given by Eq. (106) for the colorless Fredkin model and

$$
\mathcal{V}^{m_A m_B}_{hh' z_A z_B z} = \frac{\mathcal{M}^{(\ell_D)}_{0h} \mathcal{L}^{(\ell_A)}_{h(h+m_A)z_A} \mathcal{L}^{(\ell_C)}_{(h+m_A)(h'-m_B)z} \mathcal{L}^{(\ell_B)}_{(h'-m_B)h' z_B} \mathcal{M}^{(\ell_E)}_{h'0}}{\mathcal{M}^{(\ell_D + \ell_A + \ell_C + \ell_B + \ell_E)}}, \tag{177}
$$

where $\mathcal{L}_{hh'z}^{\ell}$ is given by Eq. (136) for the colorless Motzkin model, the diagonal and the off-diagonal matrix elements of $\mathbb{V}_n$, with $n = m_A + m_B = \tilde{m}_A + \tilde{m}_B$, on the basis of all possible magnetization configurations $\{(m_A, m_B)\}$, are the following

$$\mathcal{V}^{(m_A, m_B)(m_A, m_B)} = \sum_{hh'\{z\}} \mathcal{V}_{hh'z_A z_B z}^{m_A m_B}, \tag{178}$$

$$\mathcal{V}^{(m_A, m_B)(\tilde{m}_A, \tilde{m}_B)} = \sum_{hh'\{z\}} \sqrt{\mathcal{V}_{hh'z_A z_B z}^{m_A m_B} \mathcal{V}_{hh'\tilde{z}_A \tilde{z}_B z}^{\tilde{m}_A \tilde{m}_B}}. \tag{179}$$

For $\ell_A, \ell_B \ll \ell_D, \ell_E$ one verifies that the matrix elements of the blocks lose their dependence on $\ell_C$ and become

$$\mathcal{V}^{(m_A, m_B)(\tilde{m}_A, \tilde{m}_B)} \to \frac{\sqrt{D_{m_A}^{(\ell_A)} D_{m_B}^{(\ell_B)} D_{\tilde{m}_A}^{(\ell_A)} D_{\tilde{m}_B}^{(\ell_B)}}}{2^{\ell_A + \ell_B}} \tag{180}$$

for the Fredkin model, where

$$D_m^{(\ell)} = \binom{\ell}{\frac{\ell + |m|}{2}} p_{\ell+m}, \tag{181}$$

with $p_n = (1 - \mathrm{mod}(n, 2))$ which selects even integers, as introduced before, while

$$\mathcal{V}^{(m_A, m_B)(\tilde{m}_A, \tilde{m}_B)} \to \frac{\sqrt{M_{m_A}^{(\ell_A)} M_{m_B}^{(\ell_B)} M_{\tilde{m}_A}^{(\ell_A)} M_{\tilde{m}_B}^{(\ell_B)}}}{3^{\ell_A + \ell_B}} \tag{182}$$

for the Motzkin model, where

$$M_m^{(\ell)} = \sum_{k=|m|}^{\ell} \binom{k}{\frac{k + |m|}{2}} \binom{\ell}{k} p_{k+m}. \tag{183}$$

Notice that $D_m^{(\ell)} = \mathcal{D}_{h\,h+m}^{(\ell)}$, for large $h$, namely for $h > \ell$, and, analogously, $M_m^{(\ell)} = \mathcal{M}_{h\,h+m}^{(\ell)}$, for $h > \ell$. Both quantities are the number of some lattice paths starting from $(0,0)$ and ending at $(\ell, m)$ without constraints. One can verify that

$$\sum_{m=-\ell}^{\ell} D_m^{(\ell)} = 2^{\ell}, \tag{184}$$

$$\sum_{m=-s}^{\ell} M_m^{(\ell)} = 3^{\ell}, \tag{185}$$

which are the numbers of all possible configurations of $\ell$ $\frac{1}{2}$-spins and $\ell$ 1-spins, respectively. Remarkably, we find that all off-diagonal terms in each block are written in terms of the diagonal probabilities, therefore these coherent terms persist for any set of lengths. In particular, looking at Eqs. (180), (182), we notice that the blocks identifying the magnetization sectors, when deeply in the bulk, become singular matrices, since any two rows or columns are equal except for a factor. As a result only a single eigenvalue of a generic block $\mathbb{V}_n$ is not zero, therefore, it is given by

$$\mathrm{Tr}\, \mathbb{V}_n = \sum_{i=-\ell_A}^{\ell_A} \frac{D_i^{(\ell_A)} D_{n-i}^{(\ell_B)}}{2^{\ell_S}} = \frac{D_n^{(\ell_S)}}{2^{\ell_S}} \tag{186}$$

for the Fredkin case, and

$$\mathrm{Tr}\, \mathbb{V}_n = \sum_{i=-\ell_A}^{\ell_A} \frac{M_i^{(\ell_A)} M_{n-i}^{(\ell_B)}}{3^{\ell_S}} = \frac{M_n^{(\ell_S)}}{3^{\ell_S}} \tag{187}$$

for the Motzkin case. We have found, in this way, the non-zero $(\ell_S + 1)$ eigenvalues of the reduced density matrix $\rho_{AB}$ of $A \cup B$ in the deep bulk for the colorless Fredkin model and the non-zero $(2\ell_S + 1)$ eigenvalues of $\rho_{AB}$ for the colorless Motzkin model. As a result, the entropy, deeply inside the bulk, becomes

$$S_{AB} \to -\sum_{n=-\ell_S}^{\ell_S} \left[ \frac{\mathsf{D}_n^{(\ell_S)}}{2^{\ell_S}} \log\left( \frac{\mathsf{D}_n^{(\ell_S)}}{2^{\ell_S}} \right) \right] \tag{188}$$

for the Fredkin model, and

$$S_{AB} \to -\sum_{n=-\ell_S}^{\ell_S} \left[ \frac{\mathsf{M}_n^{(\ell_S)}}{3^{\ell_S}} \log\left( \frac{\mathsf{M}_n^{(\ell_S)}}{3^{\ell_S}} \right) \right] \tag{189}$$

for the Motzkin one. As we will see these entropies are the same of those obtained for a single subsystem of size $\ell_S = \ell_A + \ell_B$.

Let us consider now the reduced density matrices of the two regions $A$ and $B$, separately, which are diagonal matrices with elements obtained by tripartition

$$\rho_A = \begin{pmatrix} \mathcal{A}^{(\ell_A)} & \cdots & 0 \\ \vdots & \ddots & \vdots \\ 0 & \cdots & \mathcal{A}^{(-\ell_A)} \end{pmatrix}, \qquad \rho_B = \begin{pmatrix} \mathcal{B}^{(\ell_B)} & \cdots & 0 \\ \vdots & \ddots & \vdots \\ 0 & \cdots & \mathcal{B}^{(-\ell_B)} \end{pmatrix}, \tag{190}$$

where

$$\mathcal{A}^{(m)} = \sum_h \frac{\mathcal{D}_{0h}^{(\ell_D)} \mathcal{D}_{hh+m}^{(\ell_A)} \mathcal{D}_{h+m,0}^{(\ell_C+\ell_B+\ell_E)}}{\mathcal{D}^{(\ell_D+\ell_A+\ell_C+\ell_B+\ell_E)}}, \qquad \mathcal{B}^{(m)} = \sum_h \frac{\mathcal{D}_{0h-m}^{(\ell_D+\ell_A+\ell_C)} \mathcal{D}_{h-mh}^{(\ell_B)} \mathcal{D}_{h0}^{(\ell_E)}}{\mathcal{D}^{(\ell_D+\ell_A+\ell_C+\ell_B+\ell_E)}} \tag{191}$$

for the Fredkin model, and

$$\mathcal{A}^{(m)} = \sum_h \frac{\mathcal{M}_{0h}^{(\ell_D)} \mathcal{M}_{hh+m}^{(\ell_A)} \mathcal{M}_{h+m,0}^{(\ell_C+\ell_B+\ell_E)}}{\mathcal{M}^{(\ell_D+\ell_A+\ell_C+\ell_B+\ell_E)}}, \qquad \mathcal{B}^{(m)} = \sum_h \frac{\mathcal{M}_{0h-m}^{(\ell_D+\ell_A+\ell_C)} \mathcal{M}_{h-mh}^{(\ell_B)} \mathcal{M}_{h0}^{(\ell_E)}}{\mathcal{M}^{(\ell_D+\ell_A+\ell_C+\ell_B+\ell_E)}} \tag{192}$$

for the Motzkin model. Also in this case, for $A$ and $B$ very far apart from the edges, the coefficients become

$$\mathcal{A}^{(m)} \to \frac{\mathsf{D}_m^{(\ell_A)}}{2^{(\ell_A)}}, \qquad \mathcal{B}^{(m)} \to \frac{\mathsf{D}_m^{(\ell_B)}}{2^{(\ell_B)}} \tag{193}$$

$$\mathcal{A}^{(m)} \to \frac{\mathsf{M}_m^{(\ell_A)}}{3^{(\ell_A)}}, \qquad \mathcal{B}^{(m)} \to \frac{\mathsf{M}_m^{(\ell_B)}}{3^{(\ell_B)}} \tag{194}$$

for the Fredkin and Motzkin cases respectively. As a result the entanglement entropies of the two subsystems have the same form of Eqs. (188) and (189), namely

$$S_A \to -\sum_{n=-\ell_A}^{\ell_A} \left[ \frac{\mathsf{D}_n^{(\ell_A)}}{2^{\ell_A}} \log\left( \frac{\mathsf{D}_n^{(\ell_A)}}{2^{\ell_A}} \right) \right], \qquad S_B \to -\sum_{n=-\ell_B}^{\ell_B} \left[ \frac{\mathsf{D}_n^{(\ell_B)}}{2^{\ell_B}} \log\left( \frac{\mathsf{D}_n^{(\ell_B)}}{2^{\ell_B}} \right) \right] \tag{195}$$

for the Fredkin model, and

$$S_A \to -\sum_{n=-\ell_A}^{\ell_A} \left[ \frac{\mathsf{M}_n^{(\ell_A)}}{3^{\ell_A}} \log\left( \frac{\mathsf{M}_n^{(\ell_A)}}{3^{\ell_A}} \right) \right], \qquad S_B \to -\sum_{n=-\ell_B}^{\ell_B} \left[ \frac{\mathsf{M}_n^{(\ell_B)}}{3^{\ell_B}} \log\left( \frac{\mathsf{M}_n^{(\ell_B)}}{3^{\ell_B}} \right) \right] \tag{196}$$

**SciPost**            SciPost Phys. 7, 053 (2019)

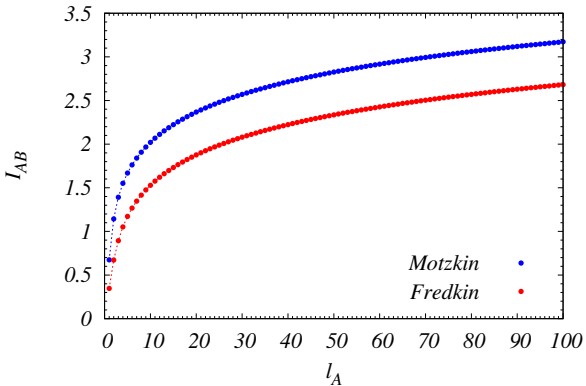

Figure 13: Mutual information between two regions of spins deep inside the bulk of a colorless Fredkin (red bottom line) and Motzkin (blue top line) chain as a function of the subsystem sizes $\ell_A = \ell_B$, obtained from Eqs. (188), (195) and Eqs. (189), (196).

for the Motzkin model. We can easily calculate the mutual information, $\mathcal{I}_{AB} = S_A + S_B - S_{AB}$, shared by two generic disjoint intervals $A$ and $B$ of sizes $\ell_A$ and $\ell_B$, from Eqs. (188) and (195) for the colorless Fredkin model and from Eqs. (189) and (196) for the colorless Motzkin model. Some results for the mutual information for the two cases with disjoint regions of same sizes, $\ell_A = \ell_B$, are plotted in Fig. 13. For $\ell_S = 2$, namely for $\ell_A = \ell_B = 1$, one recovers the results in Eqs. (134) and (167).

We have shown that the entropy for $A \cup B$ behaves as if the two regions compose a single unit subsystem of size $\ell_A + \ell_B$, no matter how far the two regions are. The effect of the off-diagonal coherent terms in the reduced density matrix is to glue together, through the central region, the two subsystems.

As a final result, for $\ell \gg 1$, approximating the binomial factors with a Gaussian distribution and the sums with the integrals, we get

$$-\sum_{n=-\ell}^{\ell} \left[ \frac{D_n^{(\ell)}}{2^\ell} \log \left( \frac{D_n^{(\ell)}}{2^\ell} \right) \right] \approx \frac{1}{2} \log \ell + \log \left( 2\sqrt{\frac{\pi}{3}} \right), \tag{197}$$

$$-\sum_{n=-\ell}^{\ell} \left[ \frac{M_n^{(\ell)}}{3^\ell} \log \left( \frac{M_n^{(\ell)}}{3^\ell} \right) \right] \approx \frac{1}{2} \log \ell + \log \left( 2\sqrt{\frac{e\,\pi}{3}} \right), \tag{198}$$

therefore, the mutual information, $\mathcal{I}_{AB} = S_A + S_B - S_{AB}$, from Eqs. (188), (195), (197) and Eqs. (189), (196), (198), becomes

$$\mathcal{I}_{AB} \approx \frac{1}{2} \log \left( \frac{\ell_A \ell_B}{\ell_A + \ell_B} \right) + \log \left( 2\sqrt{\frac{\pi}{3}} \right), \qquad \text{(Fredkin)} \tag{199}$$

$$\mathcal{I}_{AB} \approx \frac{1}{2} \log \left( \frac{\ell_A \ell_B}{\ell_A + \ell_B} \right) + \log \left( 2\sqrt{\frac{e\,\pi}{3}} \right), \quad \text{(Motzkin)} \tag{200}$$

for the Fredkin and the Motzkin spin chains respectively. These approximations are in perfect agreement with the results reported in Fig. 13. Surprisingly, the latter results for the mutual information have the same form of the logarithmic negativity and of the mutual information for conformal field theories [26] of two adjacent intervals.

# 7 Conclusions

In this paper we have shown that the ground states of the novel quantum spin models under study exhibit a robust non-local behavior, the long-distance entanglement, in addition to the violation of cluster decomposition property. Since the mutual information gives a upper bound for the squared connected correlation functions, our results are in agreement with analytical [9,23] and numerical [16] results on the long-distance behavior of some spin correlators. The strong entanglement shared by any segments of the spin chains survives at infinite distances either if the subsystems are located close to the edges or inside the bulk. This anomalous behavior has not been observed previously in the continuum version of the models [16] while we resorted to an exact calculation in order to reveal it, taking afterwards the continuum limit. This peculiar non-local behavior takes origin from the presence of off-diagonal coherent terms in the reduced density matrix which do not vanish in the thermodynamic limit, but which are missing in the continuum description. Intriguingly, we show that the mutual information of two disjoint subsystems inside the bulk of these spin chains does not depend on their distance and has the same form of the logarithmic negativity and of the mutual information for conformal field theories of two adjacent subsystems in an infinite system. This finding strengthens the belief that these models, which in spite of being described by local short-range Hamiltonians show non-local behaviors, can be promising tools for quantum information technologies.

# Acknowledgments

I thank P. Calabrese, X. Chen, E. Fradkin, W. Witczak-Krempa for useful discussions.

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
