# Peer review of "Long-distance entanglement in Motzkin and Fredkin spin chains"

_SciPost Physics, doi:SciPost Phys. 7, 053 (2019)_

## Round 1 · Referee Report · Anonymous (Referee 1) · 2019-6-11

Strengths

  1. The results are novel and give insights into the entanglement properties of an interesting family of unusual spin chain models.
  2. The introduction and summary of results are good and the motivation is well explained.
  3. There are very good explanations of the models and derivations, and the manuscript is mostly well self contained.
  4. Thorough, pedagogical treatment of the problem, with sensibly divided sections for readability.
  5. Good use of pictures to illustrate examples, and the plots are nice and clear.

Weaknesses

  1. There are very many typos and grammatical errors. In places these are enough to obfuscate the author's meaning.
  2. In a few places it isn't clear if the author is deriving something for the first time, or recapitulating a previously known result.
  3. Figure 3 is very confusing (at least to me).
  4. Some results are not discussed or interpreted, i.e. the peak in the right panel of Fig. 7.

Report

This manuscript reports on some technical results for entanglement properties of ground states in an unusual family of spin chain models. The Fredkin and Motzkin models are defined by local projective Hamiltonians for lattices of spins with special boundary conditions and three body interaction terms. There has been considerable interest in these models because they are local but have highly unconventional entanglement properties, so results that give new insights into their entanglement are useful.

The author focuses on calculating the negativity and mutual information for disjoint intervals in the ground states of these lattice models. The author shows that even when the intervals are infinitely far apart they remain entangled, and compares the case of intervals at the boundaries of the system, to those deep in the bulk. In the latter case results are found that contrast with recently proposed continuum limit models.

I think the results will be of general interest in the field of many-body entanglement and worthy of publication, but first several issues should be addressed.

  1. In the introduction the author states that for gapped systems the entanglement entropy scales like the area of the boundary between two subsystems. I believe this result is only true for ground states and possibly low lying states in the spectrum. Can the author clarify this?

  2. The author refers to 'colorful' and 'colorless' cases in the introduction, without explaining what color means in this context. Just a brief note (as in the abstract) to explain this relates to higher spin variants would avoid confusion for non-specialists.

  3. Negativity and mutual information are less familiar entanglement measures (outside of the field of quantum information) than e.g. von Neumann or Renyi entropies, and it would be useful if the author provided a very brief description of their properties and utility, to keep the manuscript self contained.

  4. I find Figure 3 really confusing. First, the notation used in the caption for the central region kets doesn't seem to match that used in Eqs. 17 and 18. Presumably this is a simplification to avoid explicit referral to all the $c,\bar{c}$ indices in A and B? Second, the many overlapping lines make it difficult to discern the individual walks. Could the author separate these up, or provide more diagrams to demonstrate e.g. the blue walks. In particular I can't understand how the highest blue point is compatible with walks that touch z=2 once - unless the very first point counts, in which case other blue paths would touch z=2 twice.

  5. At the point where they are introduced, or at Eq. 19 where the explicit form is given, it would be useful to sketch a proof and/or give citations for the relation between the coeffs $\mathcal{A}$ and the $\mathcal{D}$ numbers.

  6. The author calculates the negativity for both Fredkin and Motzkin models but I don't think there is any interpretation of the results found, nor are there plots. If the negativity is worth calculating, surely the resulting expressions are worthy of some discussion?

  7. Comparing Figure 5 and figure 7, there is a local maximum at q=3 for the mutual information between edge spins in the Motzkin model, but the result for the Fredkin case is monotonic. Can the author explain this difference and is there a physical interpretation of what is happening? I understand that the two results will differ at q=1 because the edge spins for the Fredkin chain are necessarily uncorrelated, but I don't see how this relates to the presence or absence of a peak at small q.

  8. The results for spins in the bulk of the Fredkin model are shown in Fig. 9. The mutual information as a function of distance from the edge (right panel) shows a series of steps of length two. What causes this and why is it not present for the Motzkin case, Fig 10? In addition, why not include the asymptotic values (or a best fit extrapolation of the curves) on these plots to convince the reader that they really are consistent with Eqs. 134 and 167?

  9. The author states in the intro and conclusion that the some of the results are in contradiction to those found in the continuum limit version of the model. Based on this can the author make a concrete statement as to whether the continuum limit model is actually a good representation of the Motzkin and Fredkin spin chains?

  10. Finally but importantly, there is a very large number of typos and grammatical errors throughout the entire manuscript. These serve to make the paper confusing and risk leading to misinterpretations of the author's meaning. As an example, at one point the author uses 'ones' when they mean 'once'.

Requested changes

  1. Correct spelling and grammar throughout the manuscript.
  2. Clarify the statement on entanglement scaling in gapped systems in the intro.
  3. Include a very brief description of what 'color' means in the intro.
  4. Include a brief description of the properties and meaning of entanglement negativity and mutual information (probably in the introduction).
  5. Fix or clarify the panels in Figure 3.
  6. Citation (or sketch proof if it is trivial) for Eqs. 19 and 20 (and possibly 66 and 67).
  7. When the notation $()^t$ is first used (between Eqs 32 and 33) please indicate that it means to take the 'transpose'.
  8. Add some discussion on the meaning of the results found for the negativity (and possibly plot them).
  9. Discuss the peak in the right panel of Fig 7.
  10. Discuss the staircase effect in Fig. 9 versus Fig. 10 and show the asymptotic values/extrapolations.

  • validity: high
  • significance: good
  • originality: good
  • clarity: high
  • formatting: excellent
  • grammar: below threshold

Author:  Luca Dell'Anna  on 2019-07-16  [id 565]

(in reply to Report 1 on 2019-06-11)
Category:
answer to question
correction

Dear Editor,

I would like to thank the Referee for his/her careful reading of the paper, for his/her positive assessment of the work and for his/her suggestions useful to improve the quality of the manuscript. Hereafter I resubmit a new version of the paper with corrections and additions made following the 10 Referee’s comments.

1) In the Introduction I clarified that the entanglement entropy “quantifies how the two parts of the whole system in a quantum state are entangled…. for gapped systems in the ground states, it scales with the area of the boundary of the two subsystems”

2) In the introduction, as done in the abstract, I included the meaning of “colorless” and “colorful” cases saying “…in the colorless cases (spin-1 Motzkin model and spin-1/2 Fredkin model)… This behavior has been shown to occur … for the colorful cases (for spins larger than 1)”

3) In the introduction I added the following paragraph providing a brief description of mutual information and negativity: “…The latter, also called the von Neumann mutual information, is a measure of correlation between subsystems of a quantum state and is the quantum analog of the Shannon mutual information. The negativity is also a measure of quantum entanglement, based on the positive partial transpose (PPT) criterion of separability, which provides a necessary condition for a joint density matrix ρAB of two quantum mechanical systems, A and B, to be separable [18]. If the state is separable (not entangled) the negativity is zero, therefore, if the negativity is greater than zero, the state is surely entangled, nevertheless it can be zero even if the state is entangled.”

4) In Fig. 3 I modified the symbols for the states in agreement with Eq. (18). Moreover I modified the sentence which now reads: “…classified by touching at least once, but not crossing, the horizontal lines z = 0, 1, 2.”. Also in the text, before the figure, I modified (correcting a grammar error) a sentence which now reads “those paths which touch at least once the horizontal line defined by z but never cross it.” In order to have a better understanding of the figure I also included the following sentence at the end of the section (pag.8, before Sec. 4.1): “The quantity in the brackets of Eq. (20) counts the number of just those paths which touch but not cross the level z. For example, in Fig. 3, for z = 1 the paths are only those depicted by dashed red lines.”

5) Before Eq.(19) I added the following sentence (and citations): “…which is the Schmidt number resulting from the product of the normalization factors of |P(lA)⟩ and |P(L−lA)⟩, when expressed in the canonical basis, divided by the normalization factor of |P(L)⟩ and the number of color degrees of freedom for h up-spins [9, 10]”, and before Eq. (20), “…The coefficient in Eq. (17), for the decomposition into three parts, analogously to Eq. (19), is given by the product of the normalization factors of |P(lA) 0h⟩, |P(L−lA−lB)hh′(z)⟩ and |P(lB)h′0⟩, when these states are expressed in the canonical basis, divided by the normalization factor of |P(L)⟩ and the total number of color degrees of freedom for (h+h’−z) up-spins, so that it reads…”. Before Eq.(67) I added the sentence (and citations) “…analogously to Eq. (19), are given by [7, 8]…” and before Eq.(67) “…analogously to what found for the Fredkin model in Eq. (20),…”

6) In Sec. 4.3 (pag. 10) I added the plot for the negativity in the Fredkin model (new Fig. 5) adding a paragraph for the discussion of the results: “We found, therefore, that, for q > 3, the negativity N does not vanish even at infinite distance between the two spins at the edges of the chain. This means that the quantum state in Eq. (29) is surely distantly entangled. On the other hand, for q = 1 the state is separable while for q = 2 the state described by Eq. (29) is a Werner state with zero negativity, nevertheless it is entangled, is a so-called PPT entangled state, as we will show in the next section using another entanglement measure. For q = 3 the negativity is N = 1/(L − 1), therefore the state is surely entangled for any finite L. Also in this case we will see that, even if the negativity goes to zero for infinite distance, the state in Eq. (29) is long-distance entangled, as shown by the following calculation.” In Sec. 5.3 (pag. 17) I provided a plot also for the negativity in the Motzkin case (new Fig. 8) with the following discussion: “Also for the Motzkin model we have that, for q > 3, the negativity N does not vanish even at infinite distance between the two spins at the edges of the chain. This means that the quantum state in Eq. (77) is surely distantly entangled. On the other hand, for q = 1 and q = 2 the negativity is zero, nevertheless Eq. (77) is a PPT entangled state, as we will show in the next section. For q = 3 the negativity is N ≈ 7/9L, therefore the state is surely entangled for any finite L, but also in this case we will see that, even if the negativity goes to zero for infinite distance, the state in Eq. (77) is infinitely long-distance entangled, as shown by the following calculation.”

7) Commenting Fig. 9 (previously, Fig.7), at the end of pag. 19 I added the following sentence: “The expression in Eq. (103) as a function of q perfectly agrees with the results reported in Fig. 9 (right panel) obtained calculating the asymptotic IAB for several values of q. As one can see from Fig. 9, comparing it with Fig. 6, unlike the Fredkin model, because of the greater complexity due to the spin degrees of freedom, IAB for the Motzkin model does not monotonically increase with q.” Actually Eq.(103) has been derived exactly to check the non monotonic behavior found after calculating the mutual informations for several q at large L.

8) At the end of Sec. 6.1 (pag. 23) I added the sentence “The staircase effect observed in Fig. 11 (right panel) is due to the fact that D(l)hh′ are zero for (l+h+h’) odd integers.This feature is also present in other ground-state quantities like the magnetization and the correlation functions [9, 21].” I decided not to put the asymptotic behavior on Fig. 11 (previously Fig. 9) and Fig.12 (previously, Fig. 10) further increasing l_D because the behaviors can be affected by finite size effects. However the paper contains explicitly all the analytic expressions needed to perform finite size scaling analysis. However Eqs. (134) and (167) are obtained again also from the general discussion made in Sec. 6.3, (pag. 30) when I said “…for lA = lB = 1, one recovers the results in Eqs. (134) and (167).”

9) In order to stress that the results reported in this paper is consistent with other findings, I modified a sentence in the abstract that now reads “This strongly entangled behavior, occurring both for colorful versions of the models (with spin larger than 1/2 or 1, respectively) and for colorless cases (spin 1/2 and 1), is consistent with the violation of the cluster decomposition property.” and in the conclusions: “Since the mutual information gives a upper bound for the squared connected correlation functions, our results are in agreement with analytical [9, 21] and numerical [16] results on the long-distance behavior of some spin correlators.”. In order to clarify the discrepancy with the continuum description of the models I also modified the following sentence: ” This peculiar non-local behavior takes origin from the presence of off-diagonal coherent terms in the reduced density matrix which do not vanish in the thermodynamic limit, but which are missing in the continuum description.”

10) I fixed the typos and grammar errors.

---

## Round 2 · Referee Report · Anonymous (Referee 1) · 2019-10-1

Report

I think the manuscript is improved and I am satisfied with the author's changes and responses to my requested changes.
I now recommend publication, however I did find at least one typo that a standard spellcheck would have turned up (bottom of page 7, "Before to proceed ne need to know").

---

## Round 2 · Author Response

Dear Editor,
hereafter the new version of the paper with the changes made in accordance with the requests of the Referee.
The reply and the changes have been already sent in July.

---

## Round 2 · List of Changes

The list of changes has been sent already in the reply to the Referee's report in July.

---

## Editorial Decision

published